# Contrastive Attraction and Contrastive Repulsion for Representation Learning

**Huangjie Zheng**[*]                                  *huangjie.zheng@utexas.edu*
*Department of Statistics and Data Science*
*The University of Texas at Austin*

**Xu Chen**[*]                                          *xuchen2016@sjtu.edu.cn*
*Shanghai Jiao Tong University*
*Alibaba Group*

**Jiangchao Yao**                                       *sunaker@sjtu.edu.cn*
*Cooperative Medianet Innovation Center, Shanghai Jiao Tong University*
*Shanghai AI Laboratory*

**Hongxia Yang**                                        *hongxia.yang1@gmail.com*
*Shanghai Institute for Advanced Study of Zhejiang University (SIAS)*

**Chunyuan Li**                                         *chunyuan.li@microsoft.com*
*Microsoft Research, Redmond*

**Ya Zhang**                                            *ya_zhang@sjtu.edu.cn*
*Cooperative Medianet Innovation Center, Shanghai Jiao Tong University*
*Shanghai AI Laboratory*

**Hao Zhang**                                           *zhanghao01@xidian.edu.cn*
*Xidian University*

**Ivor Tsang**                                          *ivor_tsang@cfar.a-star.edu.sg*
*A⋆STAR Centre for Frontier AI Research (CFAR)*

**Jingren Zhou**                                        *jingren.zhou@alibaba-inc.com*
*Alibaba Group*

**Mingyuan Zhou**                                       *mingyuan.zhou@mccombs.utexas.edu*
*McCombs School of Business*
*The University of Texas at Austin*

**Reviewed on OpenReview:** *https: // openreview. net/ forum? id= f39UIDkwwc*

## Abstract

Contrastive learning (CL) methods effectively learn data representations in a self-supervision manner, where the encoder contrasts each positive sample over multiple negative samples via a one-vs-many softmax cross-entropy loss. By leveraging large amounts of unlabeled image data, recent CL methods have achieved promising results when pretrained on large-scale datasets, such as ImageNet. However, most of them consider the augmented views from the same instance are positive pairs, while views from other instances are negative ones. Such binary partition insufficiently considers the relation between samples and tends to yield worse performance when generalized on images in the wild. In this paper, to further improve the performance of CL and enhance its robustness on various datasets, we propose a

---

[*]The first two authors share equal contribution

doubly CL strategy that separately compares positive and negative samples within their own groups, and then proceeds with a contrast between positive and negative groups. We realize this strategy with contrastive attraction and contrastive repulsion (CACR), which makes the query not only exert a greater force to attract more distant positive samples but also do so to repel closer negative samples. Theoretical analysis reveals that CACR generalizes CL's behavior by positive attraction and negative repulsion, and it further considers the intra-contrastive relation within the positive and negative pairs to narrow the gap between the sampled and true distribution, which is important when datasets are less curated. With our extensive experiments, CACR not only demonstrates good performance on CL benchmarks, but also shows better robustness when generalized on imbalanced image datasets. Code and pre-trained checkpoints are available at `https://github.com/JegZheng/CACR-SSL`.

# 1 Introduction

The conventional Contrastive Learning (CL) loss (Oord et al., 2018; Poole et al., 2018) has achieved remarkable success in representation learning, benefiting downstream tasks in a variety of areas (Misra & Maaten, 2020; He et al., 2020; Chen et al., 2020a; Fang & Xie, 2020; Giorgi et al., 2020). This loss typically appears in a one-vs-many softmax form to make the encoder distinguish the positive sample within multiple negative samples. In image representation learning, this scheme is widely used to encourage the encoder to learn representations that are invariant to unnecessary details in the representation space, for which the unit hypersphere is the most common assumption (Wang et al., 2017; Davidson et al., 2018; Hjelm et al., 2018; Tian et al., 2019; Bachman et al., 2019). Meanwhile, the contrast with negative samples is demystified as avoiding the collapse issue, where the encoder outputs a trivial constant, and uniformly distributing samples on the hypersphere (Wang & Isola, 2020). Beyond the usage of negative samples, several non-contrastive methods in parallel considers using momentum encoders, stop gradient operation (Caron et al., 2020; Chen & He, 2021; Chen et al., 2020a; Caron et al., 2021), *etc.*

To improve the quality of the contrast, various methods, such as large negative memory bank (Chen et al., 2020c), hard negative mining (Chuang et al., 2020; Kalantidis et al., 2020), and using strong or multi-view augmentations (Chen et al., 2020a; Tian et al., 2019; Caron et al., 2020), are proposed and succeed in learning powerful representations. Since the conventional CL loss achieves the one-vs-many contrast with a softmax cross-entropy loss, a notable concern is that the contrast could be sensitive to the sampled positive and negative pairs (Saunshi et al., 2019; Chuang et al., 2020). Given a sampled query, conventional CL methods usually randomly take one positive sample and multiple negative samples, and equally treat them in a softmax cross-entropy form, regardless of how informative they are to the query. The sampled positive pair could make the contrast either easy or difficult, while trivially selecting hard negative pairs could make the pretraining inefficient, making the pretraining become less effective when generalized to real-world data, where the labels are rarely distributed in a balanced manner (Li et al., 2020b; 2021). In recent studies, a large of negative sample manipulation is proposed to make the contrast more effective, such as ring annealing (Wu et al., 2021), maximizing margin within negatives (Shah et al., 2022), hard/soft nearest neighbor selection (Dwibedi et al., 2021; GE et al., 2023).

Considering the CL loss aims to train the encoder to distinguish the positive sample from multiple negative samples, an alternative intuition is that the positive samples need to be pulled close, while negative samples need to be pushed far away from the given query in the representation space. In addition to such a push-pull diagram, the intra-relation within positive and negative samples should also be considered. This motivates us to investigate the CL in a view of transport and propose Contrastive Attraction and Contrastive Repulsion (CACR), a doubly CL framework where the positive and negative samples are first contrasted within themselves before getting pulled and pushed from the query, respectively. As shown in Figure 1, unlike conventional CL, which equally treats samples and pulls/pushes them in the softmax cross-entropy contrast, CACR not only considers moving positive/negative samples close/away, but also models two conditional distributions to guide the movement of different samples. The conditional distributions apply a doubly-contrastive strategy to compare the positive samples and the negative ones within themselves separately. As an interpretation, if a selected positive sample is far from the query, it indicates the encoder does not sufficiently capture

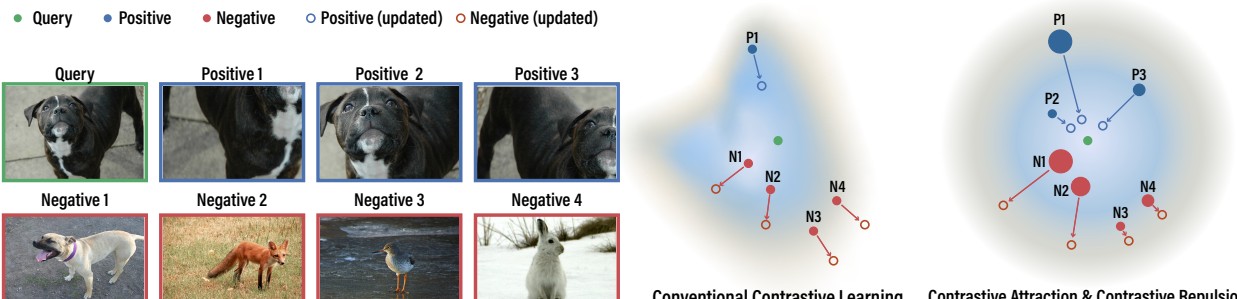

Figure 1: Comparison of conventional contrastive learning (CL) and the proposed Contrastive Attraction and Contrastive Repulsion (CACR) framework. For conventional CL, given a query, the model randomly takes one positive sample to form a positive pair and compares it against multiple negative pairs, with all samples equally treated. For CACR, using multiple positive and negative pairs, the weight of a sample (indicated by point scale) is contrastively computed to allow the query to not only more strongly pull more distant positive samples, but also more strongly push away closer negative samples.

some information. CACR will then assign a higher probability for the query to pull this positive sample. Conversely, if a selected negative sample is too close to the query, it indicates the encoder has difficulty distinguishing them, and CACR will assign a higher probability for the query to push away this negative sample. This double-contrast method contrast positive samples from negative samples, in a context of the relation within positives and negatives. We theoretically analyze CACR is universal under general situations or conditions, without the need for modification, and empirically demonstrate the learned representations are more effective and robust in various tasks. Our main contributions include:

**i)** We propose CACR, which achieves contrastive learning and produces useful representations by attracting the positive samples towards the query and repelling the negative samples away from the query, guided by two conditional distributions.

**ii)** Our theoretical analysis shows that CACR generalizes the conventional CL loss. The conditional distributions help treat the samples differently by modeling the intra-relation of positive/negative samples, which is proved to be important when the datasets are less curated.

**iii)** Our experiments demonstrate the effectiveness of CACR in a variety of standard CL settings, with both convolutional and transformer-based architectures on various benchmark datasets. Moreover, in the case where the dataset has an imbalanced label distribution, CACR has better robustness and provides consistent better pretraining results than conventional CL.

## 2    Related work

Plenty of unsupervised representation learning (Bengio et al., 2013) methods have been developed to learn good data representations, *e.g.,* PCA (Tipping & Bishop, 1999), RBM (Hinton & Salakhutdinov, 2006), VAE (Kingma & Welling, 2014). Among them, CL (Oord et al., 2018) is investigated as a lower bound of mutual information in early-stage (Gutmann & Hyvärinen, 2010; Hjelm et al., 2018). Recently, many studies reveal that the effectiveness of CL is not just attributed to the maximization of mutual information (Tschannen et al., 2019; Tian et al., 2020a). In vision tasks, SimCLR (Chen et al., 2020a;b) studies extensive augmentations for positive and negative samples and intra-batch-based negative sampling. A memory bank that caches representations (Wu et al., 2018) and a momentum update strategy are introduced to enable the use of an enormous number of negative samples (He et al., 2020; Chen et al., 2020c). Tian et al. (2019; 2020b) consider the image views in different modalities and minimize the irrelevant mutual information between them. Empirical researches observe the merits of using "hard" negative samples in CL, motivating additional techniques, such as Mixup and adversarial noise (Bose et al., 2018; Cherian & Aeron, 2020; Li et al., 2020a). CL has also been developed in learning representations for text (Logeswaran & Lee, 2018), sequential data (Oord et al., 2018; Hénaff et al., 2019), structural data like graphs (Sun et al., 2020a; Li et al., 2019; Hassani & Khasahmadi, 2020; Velickovic et al., 2019), reinforcement learning (Srinivas et al., 2020), downstream fine-tuning scenarios (Khosla et al., 2020; Sylvain et al., 2020; Cui et al., 2021). Besides vision tasks, CL

methods are widely applied to benefit in a variety of areas such as NLP, graph learning, and cross-modality learning (Misra & Maaten, 2020; He et al., 2020; Chen et al., 2020d; Fang & Xie, 2020; Giorgi et al., 2020; Gao et al., 2021; Korbar et al., 2018; Jiao et al., 2020; Li & Zhao, 2021; Monfort et al., 2021).

In a view that not all negative pairs are "true" negatives (Saunshi et al., 2019), Chuang et al. (2020) propose a decomposition of the data distribution to approximate the true negative distribution. RingCL (Wu et al., 2021) proposes to use "neither too hard nor too easy" negative samples by predefined percentiles, and HN-CL (Robinson et al., 2021) applies Monte-Carlo sampling for selecting hard negative samples. Zhang et al. selects the top-k important samples based on feature similarity. Shah et al. (2022) selects negatives as the sparse support vectors and optimize in a max-margin manner. Besides, hard/soft nearest neighbor selection are also consider as an effective way to select useful negative samples (Dwibedi et al., 2021; GE et al., 2023). Following works like Wang & Isola (2020), which reveal the contrastive scheme is optimizing the alignment of positive samples and keeping the uniformity of negative pairs, instead of abusively using negative pairs, recent self-supervised methods do not necessarily require negative pairs, avoiding the collapse issue with stop gradient or a momentum updating strategy (Chen & He, 2021; Grill et al., 2020; Caron et al., 2021). In addition, Zbontar et al. (2021) propose to train the encoder to make positive feature pairs have higher correlation and decrease the cross-correlation in different feature dimensions to avoid the collapse. Another category is based on clustering, Caron et al. (2020), Feng & Patras (2022) and Li et al. (2020c) introduce the prototypes as a proxy and train the encoder by learning to predict the cluster assignment. CACR is closely related to the previous methods, and additionally consider the relation within positive samples and negative samples. In our work, we leverage two conditional distribution to describe the relation between both positives and negatives with respect to the query samples.

## 3 The proposed approach

In CL, for observations $\boldsymbol{x}_{0:M} \sim p_{\text{data}}(\boldsymbol{x})$, we commonly assume that each $\boldsymbol{x}_i$ can be transformed in certain ways, with the samples transformed from the same and different data regarded as positive and negative samples, respectively. Specifically, we denote $\mathcal{T}(\boldsymbol{x}_i, \epsilon_i)$ as a random transformation of $\boldsymbol{x}_i$, where $\epsilon_i \sim p(\epsilon)$ represents the randomness injected into the transformation. In computer vision, $\epsilon_i$ often represents a composition of random cropping, color jitter, Gaussian blurring, *etc.* For each $\boldsymbol{x}_0$, with query $\boldsymbol{x} = \mathcal{T}(\boldsymbol{x}_0, \epsilon_0)$, we sample a positive pair $(\boldsymbol{x}, \boldsymbol{x}^+)$, where $\boldsymbol{x}^+ = \mathcal{T}(\boldsymbol{x}_0, \epsilon^+)$, and $M$ negative pairs $\{(\boldsymbol{x}, \boldsymbol{x}_i^-)\}_{1:M}$, where $\boldsymbol{x}_i^- = \mathcal{T}(\boldsymbol{x}_i, \epsilon_i^-)$. Denote $\tau \in \mathbb{R}^+$, where $\mathbb{R}^+ := \{x : x > 0\}$, as a temperature parameter. With encoder $f_{\boldsymbol{\theta}} : \mathbb{R}^n \to \mathcal{S}^{d-1}$, where we follow the convention to restrict the learned $d$-dimensional features with a unit norm, we desire to have similar and distinct representations for positive and negative pairs, respectively, via the contrastive loss as

$$\mathop{\mathbb{E}}_{(\boldsymbol{x}, \boldsymbol{x}^+, \boldsymbol{x}_{1:M}^-)} \left[ - \ln \frac{e^{f_{\boldsymbol{\theta}}(\boldsymbol{x})^\top f_{\boldsymbol{\theta}}(\boldsymbol{x}^+)/\tau}}{e^{f_{\boldsymbol{\theta}}(\boldsymbol{x})^\top f_{\boldsymbol{\theta}}(\boldsymbol{x}^+)/\tau} + \sum_i e^{f_{\boldsymbol{\theta}}(\boldsymbol{x}_i^-)^\top f_{\boldsymbol{\theta}}(\boldsymbol{x})/\tau}} \right]. \tag{1}$$

Note by construction, the positive sample $\boldsymbol{x}^+$ is independent of $\boldsymbol{x}$ given $\boldsymbol{x}_0$ and the negative samples $\boldsymbol{x}_i^-$ are independent of $\boldsymbol{x}$. Intuitively, this 1-vs-$M$ softmax cross-entropy encourages the encoder to not only pull the representation of a randomly selected positive sample closer to that of the query, but also push the representations of $M$ randomly selected negative samples away from that of the query.

### 3.1 Contrastive attraction and contrastive repulsion

In the same spirit of letting the query attract positive samples and repel negative samples, Contrastive Attraction and Contrastive Repulsion (CACR) directly models the cost of moving from the query to positive/negative samples with a doubly contrastive strategy:

$$\mathcal{L}_{\text{CACR}} := \underbrace{\mathbb{E}_{\boldsymbol{x} \sim p(\boldsymbol{x})} \mathbb{E}_{\boldsymbol{x}^+ \sim \pi_{\boldsymbol{\theta}}^+(\cdot \,|\, \boldsymbol{x}, \boldsymbol{x}_0)} \left[ c(f_{\boldsymbol{\theta}}(\boldsymbol{x}), f_{\boldsymbol{\theta}}(\boldsymbol{x}^+)) \right]}_{\textbf{Contrastive Attraction}}$$

$$+ \underbrace{\mathbb{E}_{\boldsymbol{x} \sim p(\boldsymbol{x})} \mathbb{E}_{\boldsymbol{x}^- \sim \pi_{\boldsymbol{\theta}}^-(\cdot \,|\, \boldsymbol{x})} \left[ -c(f_{\boldsymbol{\theta}}(\boldsymbol{x}), f_{\boldsymbol{\theta}}(\boldsymbol{x}^-)) \right]}_{\textbf{Contrastive Repulsion}},$$

$$:= \mathcal{L}_{\text{CA}} + \mathcal{L}_{\text{CR}}, \tag{2}$$

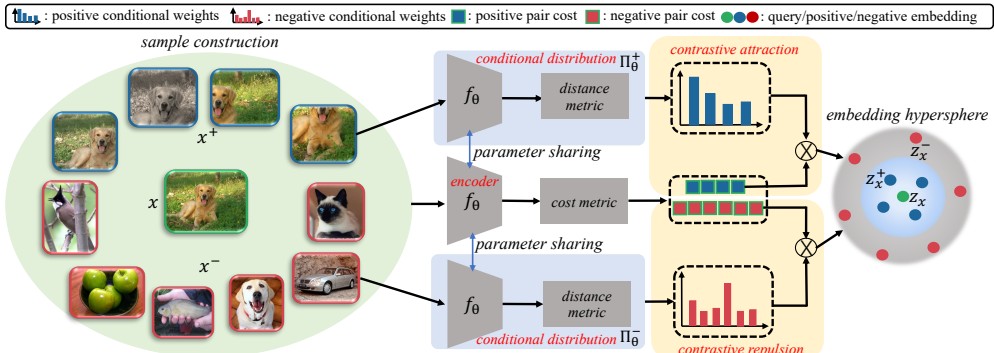

Figure 2: Illustration of the CACR framework. The encoder extracts features from samples and the conditional distributions help weigh the samples differently given the query, according to the distance of a query $\boldsymbol{x}$ and its contrastive samples $\boldsymbol{x}^+, \boldsymbol{x}^-$. $\otimes$ means element-wise multiplication between costs and conditional weights.

where we denote $\boldsymbol{\pi}^+$ and $\boldsymbol{\pi}^-$ as the conditional distributions of intra-positive contrasts and intra-negative contrasts, respectively, and $c(\boldsymbol{z}_1, \boldsymbol{z}_2)$ as the point-to-point cost of moving between two vectors $\boldsymbol{z}_1$ and $\boldsymbol{z}_2$, *e.g.*, the squared Euclidean distance $\|\boldsymbol{z}_1 - \boldsymbol{z}_2\|^2$ or the negative inner product $-\boldsymbol{z}_1^T \boldsymbol{z}_2$. In the following we explain the doubly contrastive components with more details.

**Contrastive attraction**: The intra-positive contrasts is defined in a form of the conditional probability, where the positive samples compete to gain a larger probability to be moved from the query. Here we adapt to CACR a Bayesian strategy in Zheng & Zhou (2021), which exploits the combination of an energy-based likelihood term and a prior distribution, to quantify the difference between two implicit probability distributions given their empirical samples. Specifically, denoting $d_{t^+}(\cdot, \cdot)$ as a distance metric with temperature $t^+ \in \mathbb{R}^+$, *e.g.*, $d_{t^+}(\boldsymbol{z}_1, \boldsymbol{z}_2) = t^+\|\boldsymbol{z}_1 - \boldsymbol{z}_2\|^2$, given a query $\boldsymbol{x} = \mathcal{T}(\boldsymbol{x}_0, \epsilon_0)$, we define the conditional probability for moving it to positive sample $\boldsymbol{x}^+ = \mathcal{T}(\boldsymbol{x}_0, \epsilon^+)$ as

$$\pi_{\boldsymbol{\theta}}^+(\boldsymbol{x}^+ \mid \boldsymbol{x}, \boldsymbol{x}_0) := \frac{e^{d_{t^+}(f_{\boldsymbol{\theta}}(\boldsymbol{x}), f_{\boldsymbol{\theta}}(\boldsymbol{x}^+))} p(\boldsymbol{x}^+ \mid \boldsymbol{x}_0)}{Q^+(\boldsymbol{x} \mid \boldsymbol{x}_0)}$$
$$Q^+(\boldsymbol{x} \mid \boldsymbol{x}_0) =: \int e^{d_{t^+}(f_{\boldsymbol{\theta}}(\boldsymbol{x}), f_{\boldsymbol{\theta}}(\boldsymbol{x}^+))} p(\boldsymbol{x}^+ \mid \boldsymbol{x}_0) d\boldsymbol{x}^+, \tag{3}$$

where $f_{\boldsymbol{\theta}}(\cdot)$ is an encoder parameterized by $\boldsymbol{\theta}$ and $Q^+(\boldsymbol{x})$ is a normalization term. This construction makes it more likely to pull $\boldsymbol{x}$ towards a positive sample that is more distant in their latent representation space. With equation 3, the contrastive attraction loss $\mathcal{L}_{\text{CA}}$ measures the cost of moving a query to its positive samples, as defined in equation 2, which more heavily weighs $c(f_{\boldsymbol{\theta}}(\boldsymbol{x}), f_{\boldsymbol{\theta}}(\boldsymbol{x}^+))$ if $f_{\boldsymbol{\theta}}(\boldsymbol{x})$ and $f_{\boldsymbol{\theta}}(\boldsymbol{x}^+)$ are further away from each other, providing flexible distributions in hard-positive selection (Wang et al., 2020).

**Contrastive repulsion**: On the contrary of the contrastive attraction shown in equation 3, we define the conditional probability for moving query $\boldsymbol{x}$ to a negative sample as

$$\pi_{\boldsymbol{\theta}}^-(\boldsymbol{x}^- \mid \boldsymbol{x}) := \frac{e^{-d_{t^-}(f_{\boldsymbol{\theta}}(\boldsymbol{x}), f_{\boldsymbol{\theta}}(\boldsymbol{x}^-))} p(\boldsymbol{x}^-)}{Q^-(\boldsymbol{x})},$$
$$Q^-(\boldsymbol{x}) := \int e^{-d_{t^-}(f_{\boldsymbol{\theta}}(\boldsymbol{x}), f_{\boldsymbol{\theta}}(\boldsymbol{x}^-))} p(\boldsymbol{x}^-) d\boldsymbol{x}^-, \tag{4}$$

where $t^- \in \mathbb{R}^+$ is the temperature. This construction makes it more likely to move query $\boldsymbol{x}$ to a negative sample that is closer from it in their representation space. With equation 4, the contrastive repulsion loss $\mathcal{L}_{\text{CR}}$ measures the expected cost to repel negative samples from the query shown in equation 2, which more heavily weighs $c(f_{\boldsymbol{\theta}}(\boldsymbol{x}), f_{\boldsymbol{\theta}}(\boldsymbol{x}^-))$ if $f_{\boldsymbol{\theta}}(\boldsymbol{x})$ and $f_{\boldsymbol{\theta}}(\boldsymbol{x}^-)$ are closer to each other. To this sense, the distribution $\pi_{\boldsymbol{\theta}}^-(\boldsymbol{x}^- \mid \boldsymbol{x})$ also assigns larger weights hard-negatives (Robinson et al., 2021).

**Choice of $c(\cdot, \cdot)$, $d_{t^+}(\cdot, \cdot)$ and $d_{t^-}(\cdot, \cdot)$.** There could be various choices for the point-to-point cost function $c(\cdot, \cdot)$, distance metric $d_{t^+}(\cdot, \cdot)$ in equation 3, and $d_{t^-}(\cdot, \cdot)$ in equation 4. Considering the encoder $f_{\boldsymbol{\theta}}$ outputs normalized vectors on the surface of a hypersphere, maximizing the inner product is equivalent to minimizing

Table 1: Comparison with representative CL methods. $K$ and $M$ denotes the number of positive and negative samples, respectively.

| Method | Contrast Loss | Intra-positive contrast | Intra-negative contrast |
|---|---|---|---|
| CL (Chen et al., 2020a) | 1-vs-$M$ cross-entropy | ✗ | ✗ |
| AU-CL (Wang & Isola, 2020) | 1-vs-$M$ cross-entropy | ✗ | ✗ |
| HN-CL (Robinson et al., 2021) | 1-vs-$M$ cross-entropy | ✗ | ✓ |
| CMC (Tian et al., 2019) | $\binom{K}{2} \times$ (1-vs-$M$ cross-entropy) | ✗ | ✗ |
| CACR (ours) | Intra-$K$-positive vs Intra-$M$-negative | ✓ | ✓ |

squared Euclidean distance. Without loss of generality, we define them as

$$c(\boldsymbol{x}, \boldsymbol{y}) = \|\boldsymbol{x} - \boldsymbol{y}\|_2^2$$
$$d_{t^+}(\boldsymbol{x}, \boldsymbol{y}) = t^+ \|\boldsymbol{x} - \boldsymbol{y}\|_2^2; \ t^+ \in \mathbb{R}_+,$$
$$d_{t^-}(\boldsymbol{x}, \boldsymbol{y}) = t^- \|\boldsymbol{x} - \boldsymbol{y}\|_2^2; \ t^- \in \mathbb{R}_+.$$

where $t^+, t^- \in \mathbb{R}^+$. There are other choices for $c(\cdot, \cdot)$ and we show the ablation study in Appendix B.5.

### 3.2  Mini-batch based stochastic optimization

Under the CACR loss as in equation 2, to make the learning of $f_{\boldsymbol{\theta}}(\cdot)$ amenable to mini-batch stochastic gradient descent (SGD) based optimization, we draw $(\boldsymbol{x}_i^{\text{data}}, \epsilon_i) \sim p_{data}(\boldsymbol{x})p(\epsilon)$ for $i = 1, \ldots, M$ and then approximate the distribution of the query using an empirical distribution of $M$ samples as

$$\hat{p}(\boldsymbol{x}) = \tfrac{1}{M} \sum_{i=1}^M \delta(\boldsymbol{x} - \boldsymbol{x}_i); \ \boldsymbol{x}_i = \mathcal{T}(\boldsymbol{x}_i^{\text{data}}, \epsilon_i).$$

where the $\delta(\cdot)$ denotes the Dirac delta function, and we note $\delta_{\boldsymbol{x}_i}$ as the Dirac function centered at $\boldsymbol{x}_i$ i.e., $\delta_{\boldsymbol{x}_i} = \delta(\boldsymbol{x} - \boldsymbol{x}_i)$. With query $\boldsymbol{x}_i$ and $\epsilon_{1:K} \overset{iid}{\sim} p(\epsilon)$, we approximate $p(\boldsymbol{x}_i^-)$ for equation 4 and $p(\boldsymbol{x}_i^+ \,|\, \boldsymbol{x}_i^{\text{data}})$ for equation 3  with $\boldsymbol{x}_{ik}^+ = \mathcal{T}(\boldsymbol{x}_i^{\text{data}}, \epsilon_k)$:

$$\hat{p}(\boldsymbol{x}_i^-) = \tfrac{1}{M-1} \sum_{j\neq i} \delta_{\boldsymbol{x}_j}, \quad \hat{p}(\boldsymbol{x}_i^+ \,|\, \boldsymbol{x}_i^{\text{data}}) = \tfrac{1}{K} \sum_{k=1}^K \delta_{\boldsymbol{x}_{ik}^+}. \tag{5}$$

Note we may improve the accuracy of $\hat{p}(\boldsymbol{x}_i^-)$ in equation 5 by adding previous queries into the support of this empirical distribution. Other more sophisticated ways to construct negative samples (Oord et al., 2018; He et al., 2020; Khosla et al., 2020) could also be adopted to define $\hat{p}(\boldsymbol{x}_i^-)$. We will elaborate these points when describing experiments.

Plugging equation 5 into equation 3 and equation 4, we approximate the conditional distributions with discrete distributions and obtain a mini-batch based CACR loss as $\hat{\mathcal{L}}_{\text{CACR}} = \hat{\mathcal{L}}_{\text{CA}} + \hat{\mathcal{L}}_{\text{CR}}$, where

$$\hat{\mathcal{L}}_{\text{CA}} := \tfrac{1}{M} \sum_{i=1}^M \sum_{k=1}^K \frac{e^{d_{t^+}(f_{\boldsymbol{\theta}}(\boldsymbol{x}_i), f_{\boldsymbol{\theta}}(\boldsymbol{x}_{ik}^+))}}{\sum_{k'=1}^K e^{d_{t^+}(f_{\boldsymbol{\theta}}(\boldsymbol{x}_i), f_{\boldsymbol{\theta}}(\boldsymbol{x}_{ik'}^+))}} \times c(f_{\boldsymbol{\theta}}(\boldsymbol{x}_i), f_{\boldsymbol{\theta}}(\boldsymbol{x}_{ik}^+)),$$

$$\hat{\mathcal{L}}_{\text{CR}} := -\tfrac{1}{M} \sum_{i=1}^M \sum_{j\neq i} \frac{e^{-d_{t^-}(f_{\boldsymbol{\theta}}(\boldsymbol{x}_i), f_{\boldsymbol{\theta}}(\boldsymbol{x}_j))}}{\sum_{j'\neq i} e^{-d_{t^-}(f_{\boldsymbol{\theta}}(\boldsymbol{x}_i), f_{\boldsymbol{\theta}}(\boldsymbol{x}_{j'}))}} \times c(f_{\boldsymbol{\theta}}(\boldsymbol{x}_i), f_{\boldsymbol{\theta}}(\boldsymbol{x}_j)).$$

We optimize $\boldsymbol{\theta}$ via SGD using $\nabla_{\boldsymbol{\theta}} \hat{\mathcal{L}}_{\text{CACR}}$, with the framework instantiated as in Figure 2.

### 3.3  Relation with typical CL loss

As shown in equation 2, with both the contrastive attraction component and contrastive repulsion component, CACR loss shares the same intuition of conventional CL (Oord et al., 2018; Chen et al., 2020a) in pulling positive samples closer to and pushing negative samples away from the query in their representation space. However, CACR realizes this intuition by introducing the double-contrast strategy on the point-to-point

moving cost, where the contrasts appear in the intra-comparison within positive and negative samples, respectively. The use of the double-contrast strategy clearly differs the CACR loss in equation 2 from the conventional CL loss in equation 1, which typically relies on a softmax-based contrast formed with a single positive sample and multiple equally-weighted independent negative samples. The conditional distributions in CA and CR loss also provide a more flexible way to deal with hard-positive/negative samples (Robinson et al., 2021; Wang et al., 2020; 2019; Tabassum et al., 2022; Xu et al., 2022) and does not require heavy labor in tuning the hyper-parameters for the model. A summary of the comparison between some representative CL losses and CACR is shown in Table 1.

## 4 Property analysis of CACR

### 4.1 On contrastive attraction

We first analyze the effects *w.r.t.* the positive samples. With contrastive attraction, the property below suggests that the optimal encoder produces representations invariant to the noisy details.

*Property* 1. The contrastive attraction loss $\mathcal{L}_{\mathrm{CA}}$ is optimized if and only if all positive samples of a query share the same representation as that query. More specifically, for query $\boldsymbol{x}$ that is transformed from $\boldsymbol{x}_0 \sim p_{data}(\boldsymbol{x})$, its positive samples share the same representation with it, which means

$$f_{\boldsymbol{\theta}}(\boldsymbol{x}^+) = f_{\boldsymbol{\theta}}(\boldsymbol{x}) \ \text{ for any } \ \boldsymbol{x}^+ \sim \boldsymbol{\pi}(\boldsymbol{x}^+ \,|\, \boldsymbol{x}, \boldsymbol{x}_0). \tag{6}$$

This property coincides with the characteristic (learning invariant representation) of the CL loss in Wang & Isola (2020) when achieving the optima. However, the optimization dynamic in contrastive attraction evolves in the context of $\boldsymbol{x}^+ \sim \boldsymbol{\pi}_{\boldsymbol{\theta}}(\boldsymbol{x}^+ \,|\, \boldsymbol{x}, \boldsymbol{x}_0)$, which is different from that in the CL.

**Lemma 4.1.** *Let us instantiate* $c(f_{\boldsymbol{\theta}}(\boldsymbol{x}), f_{\boldsymbol{\theta}}(\boldsymbol{x}^+)) = -f_{\boldsymbol{\theta}}(\boldsymbol{x})^\top f_{\boldsymbol{\theta}}(\boldsymbol{x}^+)$. *Then, the contrastive attraction loss* $\mathcal{L}_{\mathrm{CA}}$ *in equation 2 can be re-written as*

$$\mathbb{E}_{\boldsymbol{x}_0} \mathbb{E}_{\boldsymbol{x}, \boldsymbol{x}^+ \sim p(\cdot \,|\, \boldsymbol{x}_0)} \left[ -f_{\boldsymbol{\theta}}(\boldsymbol{x})^\top f_{\boldsymbol{\theta}}(\boldsymbol{x}^+) \frac{\pi_{\boldsymbol{\theta}}^+(\boldsymbol{x}^+ \,|\, \boldsymbol{x}, \boldsymbol{x}_0)}{p(\boldsymbol{x}^+ \,|\, \boldsymbol{x}_0)} \right],$$

*which could further reduce to the alignment loss* $\mathbb{E}_{\boldsymbol{x}_0 \sim p_{data}(\boldsymbol{x})} \mathbb{E}_{\boldsymbol{x}, \boldsymbol{x}^+ \sim p(\cdot \,|\, \boldsymbol{x}_0)} \left[ -f_{\boldsymbol{\theta}}(\boldsymbol{x})^\top f_{\boldsymbol{\theta}}(\boldsymbol{x}^+)) \right]$ *in Wang & Isola (2020), iff* $\pi_{\boldsymbol{\theta}}^+(\boldsymbol{x}^+ \,|\, \boldsymbol{x}, \boldsymbol{x}_0) = p(\boldsymbol{x}^+ \,|\, \boldsymbol{x}_0)$.

Property 1 and Lemma 4.1 jointly show contrastive attraction in CACR and the alignment loss in CL reach the same optima, while working in different sampling mechanism. In practice $\boldsymbol{x}^+$ and $\boldsymbol{x}$ are usually independently sampled augmentations in a mini-batch, as shown in Section 3.2, which raises a gap between the empirical distribution and the true distribution. Our method makes the alignment more efficient by considering the intra-relation of these positive samples to the query.

### 4.2 On contrastive repulsion

Next we analyze the effects *w.r.t.* the contribution of negative samples. Wang & Isola (2020) reveal that a perfect encoder will uniformly distribute samples on a hypersphere under an uniform isometric assumption, *i.e.*, for any uniformly sampled $\boldsymbol{x}, \boldsymbol{x}^- \overset{iid}{\sim} p(\boldsymbol{x})$, their latent representations $\boldsymbol{z} = f_{\boldsymbol{\theta}}(\boldsymbol{x})$ and $\boldsymbol{z}^- = f_{\boldsymbol{\theta}}(\boldsymbol{x}^-)$ also satisfy $p(\boldsymbol{z}) = p(\boldsymbol{z}^-)$. We follow their assumption to analyze contrastive repulsion via the following lemma.

**Lemma 4.2.** *Without loss of generality, we define the moving cost and metric in the conditional distribution as* $c(\boldsymbol{z}_1, \boldsymbol{z}_2) = d(\boldsymbol{z}_1, \boldsymbol{z}_2) = \|\boldsymbol{z}_1 - \boldsymbol{z}_2\|_2^2$. *When we are with an uniform prior, namely* $p(\boldsymbol{x}) = p(\boldsymbol{x}^-)$ *for any* $\boldsymbol{x}, \boldsymbol{x}^- \overset{iid}{\sim} p(\boldsymbol{x})$ *and* $p(\boldsymbol{z}) = p(\boldsymbol{z}^-)$ *given their latent representations* $\boldsymbol{z} = f_{\boldsymbol{\theta}}(\boldsymbol{x})$ *and* $\boldsymbol{z}^- = f_{\boldsymbol{\theta}}(\boldsymbol{x}^-)$, *then optimizing* $\boldsymbol{\theta}$ *with* $\mathcal{L}_{\mathrm{CR}}$ *in equation 2 is the same as optimizing* $\boldsymbol{\theta}$ *to minimize the mutual information between* $\boldsymbol{x}$ *and* $\boldsymbol{x}^-$:

$$I(X; X^-) = \mathbb{E}_{\boldsymbol{x} \sim p(\boldsymbol{x})} \mathbb{E}_{\boldsymbol{x}^- \sim \pi_{\boldsymbol{\theta}}^-(\cdot \,|\, \boldsymbol{x})} \left[ \ln \frac{\pi_{\boldsymbol{\theta}}^-(\boldsymbol{x}^- \,|\, \boldsymbol{x})}{p(\boldsymbol{x}^-)} \right], \tag{7}$$

*and is also the same as optimizing $\boldsymbol{\theta}$ to maximize the conditional differential entropy of $\boldsymbol{x}^-$ given $\boldsymbol{x}$:*

$$\mathcal{H}(X^- \mid X) = \mathbb{E}_{\boldsymbol{x} \sim p(\boldsymbol{x})} \mathbb{E}_{\boldsymbol{x}^- \sim \pi_{\boldsymbol{\theta}}^- (\cdot \mid \boldsymbol{x})} [-\ln \pi_{\boldsymbol{\theta}}^- (\boldsymbol{x}^- \mid \boldsymbol{x})]. \tag{8}$$

*Here the minimizer $\boldsymbol{\theta}^\star$ of $\mathcal{L}_{\mathrm{CR}}$ is also that of $I(X; X^-)$, whose global minimum zero is attained iff $X$ and $X^-$ are independent, and the equivalent maximum of $\mathcal{H}(X^- \mid X)$ indicates the optimization of $\mathcal{L}_{\mathrm{CR}}$ is essentially aimed towards the uniformity of representation about negative samples.*

We notice that one way to reach the optimum suggested in the above lemma is optimizing $\boldsymbol{\theta}$ by contrastive repulsion until that for any $\boldsymbol{x} \sim p(\boldsymbol{x})$, $d(f_{\boldsymbol{\theta}}(\boldsymbol{x}), f_{\boldsymbol{\theta}}(\boldsymbol{x}^-))$ is equal for all $\boldsymbol{x}^- \sim \pi_{\boldsymbol{\theta}}^- (\cdot \mid \boldsymbol{x})$. This means for any sampled negative samples, their representations are also uniformly distributed after contrastive repulsion. Interestingly, this is consistent with the uniformity property achieved by CL (Wang & Isola, 2020), which connects contrastive repulsion with CL in the perspective of negative sample effects.

Note that, although the above analysis builds upon the uniform isometric assumption, our method actually does not rely on it. Here, we formalize a more general relation between the contrastive repulsion and the contribution of negative samples in CL without this assumption as follows.

**Lemma 4.3.** *As the number of negative samples $M$ goes to infinity, the contribution of the negative samples to the CL loss becomes the Uniformity Loss in AU-CL (Wang & Isola, 2020), termed as $\mathcal{L}_{\mathrm{uniform}}$ for simplicity. It can be expressed as an upper bound of $\mathcal{L}_{\mathrm{CR}}$ by adding the mutual information $I(X; X^-)$ in equation 7:*

$$\underbrace{\mathbb{E}_{\boldsymbol{x} \sim p(\boldsymbol{x})} \left[ \ln \mathbb{E}_{\boldsymbol{x}^- \sim p(\boldsymbol{x}^-)} e^{f_{\boldsymbol{\theta}}(\boldsymbol{x}^-)^\top f_{\boldsymbol{\theta}}(\boldsymbol{x})/\tau} \right]}_{\mathcal{L}_{\mathrm{uniform}}} + I(X; X^-) \geqslant \mathcal{L}_{\mathrm{CR}},$$

As shown in Lemma 4.3, the mutual information $I(X; X^-)$ helps quantify the difference between $\mathcal{L}_{\mathrm{uniform}}$ and $\mathcal{L}_{\mathrm{CR}}$. The difference between drawing $\boldsymbol{x}^- \sim \pi_{\boldsymbol{\theta}}^- (\boldsymbol{x}^- \mid \boldsymbol{x})$ (in CR) and drawing $\boldsymbol{x}^-$ independently in a mini-batch (in CL) is non-trivial as long as $I(X; X^-)$ is non-zero. In practice, this is true almost everywhere since we have to handle the skewed data distribution in real-world applications, *e.g.*, the label-shift scenarios (Garg et al., 2020). In this view, CR does not require the representation space to be uniform like CL does, and is more robust to the complex cases through considering the intra-contrastive relation within negative samples.

## 5 Experiments and empirical analysis

In this section, we first study the CACR behaviors with small-scale experiments, where we use CIFAR-10, CIFAR-100 (Hinton, 2007) and create two class-imbalanced CIFAR datasets as empirical verification of our theoretical analysis. We mainly compare with representative CL methods, divided into two different categories according to their positive sampling size: $K = 1$ and $K = 4$. For methods with a single positive sample ($K = 1$), the baseline methods include the conventional CL loss (Oord et al., 2018), AlignUniform CL loss (AU-CL) (Wang & Isola, 2020), and the CL loss with hard negative sampling (HN-CL) (Robinson et al., 2021). In the case of $K = 4$, we take contrastive multi-view coding (CMC) loss (Tian et al., 2019) (align with our augmentation settings and use augmentation views instead of channels) as the comparison baseline. For a fair comparison, we keep for all methods with the same experiment setting including learning-rate, training epochs, *etc.*, but use their best temperature parameters; the mini-batch size for $K = 4$ is divided by 4 from those when $K = 1$ to make sure the encoder leverages same samples in each iteration.

For large-scale datasets, we use ImageNet-1K (Deng et al., 2009) and compare with the state-of-the-art frameworks (He et al., 2020; Zbontar et al., 2021; Chen et al., 2020a; Caron et al., 2020; Grill et al., 2020; Huynh et al., 2020) on linear probing, where we report the Top-1 validation accuracy on ImageNet-1K data. We also report the results of object detection/segmentaion following the transfer learning protocol. To further justify our analysis, we also leverage two large-scale but label-imbalanced datasets (Webvision v1 and ImageNet-22K) for linear probing pretraining. The reported numbers for baselines are from the original papers if available, otherwise we report the best ones reproduced with the settings according to their corresponding papers. Please refer to Appendix C for detailed experiment setups.

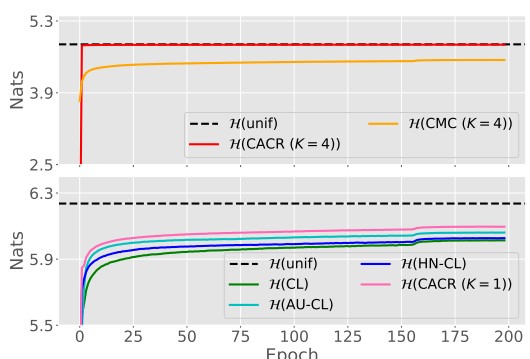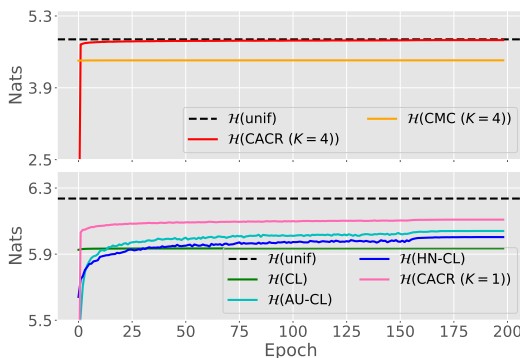

Figure 3: Conditional entropy $\mathcal{H}(X^-|X)$ *w.r.t.* epoch on CIFAR-10 (**left**) and linearly label-imbalanced CIFAR-10 (**right**). The maximal possible conditional entropy is marked by a dotted line.

### 5.1 Studies and analysis on small-scale datasets

**Classification accuracy:** To facilitate the analysis, we apply all methods with an AlexNet-based encoder following the setting in Wang & Isola (2020), trained in 200 epochs. We pretrained the encoder on regular CIFAR-10/100 data and create class-imbalanced cases by randomly sampling a certain number of samples from each class with a "linear" or "exponential" rule by following the setting in Kim et al. (2020). Specifically, given a dataset with $C$ classes, for class $l \in \{1, 2, ..., C\}$, we randomly take samples with proportion $\lfloor \frac{l}{C} \rfloor$ for "linear" rule and proportion $\exp(\lfloor \frac{l}{C} \rfloor)$ for "exponential" rule. For evaluation we keep the standard validation/testing datasets. Thus there is a label-shift between the training and testing data distributions.

Summarized in Table 2 are the results on both regular and class-imbalanced datasets. The first two columns show the results pretrained with curated data, where we can observe that in the case of $K = 1$, where the intra-positive contrast of CACR degenerates, CACR slightly outperforms all CL methods. When $K = 4$, it is interesting to observe an obvious boost in performance, where CMC improves CL by around 2-3% while CACR improves CL by around 3-4%, which supports our analysis that CA is helpful when the intra-positive contrast is not degenerated. The right four columns present the linear probing results pretrained with class-imbalanced data, which show all the methods have a performance drop. It is clear that CACR has the least performance decline in most cases. Especially, when $K = 4$, CACR shows better performance robustness due to the characteristic of doubly contrastive within positive and negative samples. For example, in the "exponential" setting of CIFAR-100, CL and HN-CL drop 12.57% and 10.73%, respectively, while CACR ($K = 4$) drops 9.24%. It is also interesting to observe HN-CL is relatively better among the baseline methods. According to Robinson et al. (2021), in HN-CL the negative samples are sampled according to the "hardness" *w.r.t.* the query samples with an intra-negative contrast. Its loss could converge to CACR ($K = 1$) with infinite negative samples. This performance gap indicates that directly optimizing the CACR loss could be superior when we have a limited number of samples. With this class-imbalanced datasets, we provide the empirical support to our analysis: When the condition in Lemma 4.2 is violated, CACR shows a clearer difference than CL and a better robustness with its unique doubly contrastive strategy within positive and negative samples.

**On the effect of CA and CR:** To further study the contrasts within positive and negative samples, in each epoch, we calculate the conditional entropy with equation 8 on every mini-batch of the *validation data* and take the average across mini-batches. Then, we illustrate in Figure 3 the evolution of conditional entropy $\mathcal{H}(X^-|X)$ *w.r.t.* the training epoch on regular CIFAR-10 and class-imbalanced CIFAR-10. As shown, $\mathcal{H}(X^-|X)$ is getting maximized as the encoder is getting optimized, indicating the encoder learns to distinguish the negative samples from given query. It is also interesting to observe that in the case with multiple positive samples, this process is much more efficient, where the conditional entropy reaches the possible biggest value rapidly. This implies the CA module can further boost the repulsion of negative samples. From the gap between CACR and CMC, we can learn although CMC uses multiple positive in CL loss, the lack of intra-positive contrast shows the gap of attraction efficiency. In the right panel of Figure 3, the difference between CACR and baseline methods are more obvious, where we can find the conditional entropy

Table 2: The linear classification accuracy (%) of different contrastive objectives on small-scale datasets, pretrained on regular and label-imbalanced CIFAR10/100 with AlexNet backbone. "Linear" and "Exponential" indicate the number of samples in each class are chosen by following a linear rule or an exponential rule, respectively. The performance drops compared with the performance in regular CIFAR data are shown next to each result.

| Label imbalance | Regular | | Linear | | Exponential | |
|---|---|---|---|---|---|---|
| Dataset | CIFAR-10 | CIFAR-100 | CIFAR-10 | CIFAR-100 | CIFAR-10 | CIFAR-100 |
| SimCLR (CL) | 83.47 | 55.41 | $79.88_{3.59\downarrow}$ | $52.29_{3.57\downarrow}$ | $71.74_{11.73\downarrow}$ | $43.29_{12.57\downarrow}$ |
| AU-CL | 83.49 | 55.31 | $80.25_{3.14\downarrow}$ | $52.74_{2.57\downarrow}$ | $71.62_{11.76\downarrow}$ | $44.38_{10.93\downarrow}$ |
| HN-CL | 83.67 | 55.87 | $\mathbf{80.51}_{3.15\downarrow}$ | $52.72_{3.14\downarrow}$ | $72.74_{10.93\downarrow}$ | $45.13_{10.73\downarrow}$ |
| CACR ($K=1$) | **83.73** | **56.52** | $80.46_{3.27\downarrow}$ | $\mathbf{54.12}_{2.40\downarrow}$ | $\mathbf{73.02}_{10.71\downarrow}$ | $\mathbf{46.59}_{9.93\downarrow}$ |
| CMC ($K=4$) | 85.54 | 58.64 | $82.20_{3.34\downarrow}$ | $55.38_{3.26\downarrow}$ | $74.77_{10.77\downarrow}$ | $48.87_{9.77\downarrow}$ |
| CACR ($K=4$) | **86.54** | **59.41** | $\mathbf{83.62}_{2.92\downarrow}$ | $\mathbf{56.91}_{2.50\downarrow}$ | $\mathbf{75.89}_{10.65\downarrow}$ | $\mathbf{50.17}_{9.24\downarrow}$ |

Table 3: The top-1 classification accuracy (%) of different contrastive objectives with different training epochs on small-scale datasets, following SimCLR setting and applying the AlexNet-based encoder.

| Dataset | Trained with 400 epochs | | | | Trained with 200 epochs | |
|---|---|---|---|---|---|---|
| | CL | AU-CL | HN-CL | CACR(K=1) | CMC(K=4) | CACR(K=4) |
| CIFAR-10 | 83.61 | 83.57 | 83.72 | **83.86** | 85.54 | **86.54** |
| CIFAR-100 | 55.41 | 56.07 | 55.80 | **56.41** | 58.64 | **59.41** |
| STL-10 | 83.49 | 83.43 | 82.41 | **84.56** | 84.50 | **85.59** |

of baselines is slightly lower than pretrained with regular CIFAR-10 data. Especially for vanilla CL loss, we can observe the conditional entropy has a slight decreasing tendency, indicating the encoder hardly learns to distinguish negative samples in this case. Conversely, CACR still shows to remain the conditional entropy at a higher level, which explains the robustness shown in Table 2, and indicating a superior learning efficiency of CACR. See Appendix B.2 for similar observations on CIFAR-100 and exponential label-imbalanced cases. In that part, we provide more quantitative and qualitative studies on the effects of conditional distributions.

**Does CACR($K \geqslant 2$) outperform by seeing more samples?** To address this concern, in our main paper, we intentionally decrease the mini-batch size as $M = 128$. Thus the total number of samples used per iteration is not greater than those used when $K = 1$. To further justify if the performance boost comes from seeing more samples when using multiple positive pairs, we also let the methods allowing single positive pair train with double epochs. As shown in Table 3, we can observe even trained with 400 epochs, the performance of methods using single positive pair still have a gap from those using multiple positive pairs.

### 5.2 Experiments on large-scale datasets

For large-scale experiments, we follow the self-supervised evaluation pipeline to examine the performance of CACR: we first leverage MoCov2 Chen et al. (2020c) design to pre-train a ResNet-50 with CACR loss, and then evaluate the capacity of the pre-trained model in a variety of tasks, including linear probing, downstream few/full-shot image classification, and detection/segmentation. Besides these tasks, additional ablation studies are provided in Appendix B.

**Linear probing:** Table 4 summarizes the results of linear classification, where a linear classifier is trained on ImageNet-1K on top of fixed representations of the pretrained ResNet50 encoder. Similar to the case on small-scale datasets, CACR consistently shows better performance than the baselines using contrastive loss, improving SimCLR and MoCov2 by 2.7% and 2.2% respectively. Compared with other non-contrastive self-supervised SOTAs, CACR also shows on par performance.

**Label-imbalanced case:** To strengthen our analysis on small-scale label-imbalanced data, we specially deploy two real-world, but less curated datasets Webvision v1 and ImageNet-22K that have long-tail label distributions for encoder pretraining and evaluate the linear classification accuracy on ImageNet-1K. We pretrain encoder with 100/20 epochs on Webvision v1/ImageNet-22K and compare with the encoder pretrained with 200 epochs on ImageNet-1K to make sure similar samples have been seen in the pretraining. The results are shown in Table 5, where we can see CACR still outperforms the MoCov2 baseline and shows better robustness when generalized to wild image data.

Table 4: Top-1 classification accuracy (%) comparison with SOTAs including non-contrastive and contrastive methods, pretrained with ResNet50 encoder on ImageNet-1K dataset. We mark Top-3 best results in bold and highlight CL methods.

| | Methods | Batch-size | Accuracy |
|---|---|---|---|
| Non-Contrastive | BarlowTwins | 1024 | 73.2 |
| | Simsiam | 256 | 71.3 |
| (wo. Negatives) | SWAV (wo/w multi-crop) | 4096 | 71.8 / **75.3** |
| | BYOL | 4096 | 74.3 |
| Contrastive | SimCLR | 4096 | 71.7 |
| | MoCov2 | 256 | 72.2 |
| | ASCL | 256 | 71.5 |
| | FNC (w multi-crop) | 4096 | **74.4** |
| | ADACLR | 4096 | 72.3 |
| (w. Negatives) | CACR (K=1) | 256 | 73.7 |
| | CACR (K=4) | 256 | **74.7** |

Table 5: Top-1 classification accuracy (%) on ImageNet-1K, with the pre-trained ResNet50 on large-scale regular (200 epochs) and label-imbalanced (100/20 epochs) datasets. The performance drops are shown next to each result.

| Pretrained data | Methods | Accuracy |
|---|---|---|
| ImageNet-1K | MoCov2 | 67.5 |
| | CACR (K=1) | 69.5 |
| | CACR (K=4) | **70.4** |
| Webvision v1 | MoCov2 | $62.3_{5.2\downarrow}$ |
| | CACR (K=1) | $64.5_{5.0\downarrow}$ |
| | CACR (K=4) | $\mathbf{66.1_{4.3\downarrow}}$ |
| ImageNet-22K | MoCov2 | $59.9_{7.6\downarrow}$ |
| | CACR (K=1) | $61.9_{7.6\downarrow}$ |
| | CACR (K=4) | $\mathbf{64.5_{5.9\downarrow}}$ |

| | Dataset | Caltech101 | CIFAR10 | CIFAR100 | Country211 | DescriTextures | EuroSAT | FER2013 | FGVC Aircraft | Food101 | GTSRB | HatefulMemes | KITTI | MNIST | Oxford Flowers | Oxford Pets | PatchCamelyon | Rendered SST2 | RESISC45 | Stanford Cars | VOC2007 | Mean Acc. | # Wins |
|---|---|---|---|---|---|---|---|---|---|---|---|---|---|---|---|---|---|---|---|---|---|---|---|
| **5-FT** | MoCov3 | 73.7 | 70.3 | 17.4 | 2.3 | 45.6 | 60.0 | 13.5 | 7.2 | 27.6 | 16.5 | 50.8 | 43.5 | 18.1 | 65.7 | 77.1 | 50.9 | 50.7 | 58.2 | 11.2 | 25.7 | 39.3 | 4 |
| | CACR | 84.8 | 67.6 | 24.3 | 2.5 | 51.2 | 73.6 | 23.0 | 21.4 | 17.0 | 23.7 | 51.8 | 45.4 | 44.0 | 81.1 | 79.4 | 58.4 | 51.2 | 49.1 | 10.8 | 69.0 | 46.5 | 16 |
| | Gains | +11.1 | -2.7 | +6.9 | +0.2 | +5.6 | +13.6 | +9.5 | +14.2 | -10.6 | +7.2 | +1.0 | +1.9 | +25.9 | +15.4 | +2.3 | +7.5 | +0.4 | -9.1 | -0.3 | +43.3 | +7.2 | |
| **5-LP** | MoCov3 | 80.8 | 78.5 | 60.5 | 4.8 | 57.1 | 77.1 | 20.5 | 11.8 | 36.6 | 31.4 | 50.7 | 46.7 | 64.1 | 79.5 | 76.2 | 54.7 | 50.0 | 61.1 | 13.4 | 47.9 | 50.2 | 4 |
| | CACR | 79.3 | 85.4 | 62.9 | 4.7 | 57.1 | 76.1 | 18.3 | 21.6 | 40.9 | 32.9 | 50.9 | 50.3 | 69.2 | 84.6 | 81.2 | 56.9 | 51.8 | 61.7 | 21.1 | 74.4 | 54.1 | 15 |
| | Gains | -1.5 | +6.9 | +2.4 | -0.1 | +0.0 | -1.0 | -2.2 | +9.8 | +4.3 | +1.5 | +0.2 | +3.6 | +5.1 | +5.1 | +5.0 | +2.2 | +1.8 | +0.6 | +7.7 | +26.5 | +3.9 | |
| **Full-FT** | MoCov3 | 93.3 | 98.1 | 88.7 | 11.7 | 71.3 | 97.3 | 68.3 | 51.9 | 84.1 | 98.8 | 54.5 | 80.5 | 99.6 | 87.1 | 90.9 | 91.4 | 52.5 | 88.6 | 67.9 | 77.6 | 77.7 | 3 |
| | CACR | 93.3 | 98.1 | 89.9 | 12.9 | 72.0 | 97.7 | 68.3 | 56.3 | 85.2 | 99.1 | 54.8 | 80.6 | 99.1 | 89.3 | 91.6 | 88.1 | 56.6 | 88.8 | 79.1 | 75.4 | 78.8 | 15 |
| | Gains | +0.0 | +0.0 | +1.2 | +1.2 | +0.7 | +0.4 | +0.0 | +4.4 | +1.1 | +0.3 | +0.3 | +0.1 | -0.5 | +2.2 | +0.7 | -3.3 | +4.1 | +0.2 | +11.2 | -2.2 | +1.1 | |
| **Full-LP** | MoCov3 | 92.1 | 96.9 | 85.3 | 13.7 | 73.1 | 95.9 | 60.1 | 48.0 | 78.0 | 78.7 | 53.7 | 68.8 | 98.4 | 89.5 | 91.4 | 86.7 | 57.1 | 86.3 | 63.0 | 81.7 | 74.9 | 8 |
| | CACR | 92.9 | 96.9 | 85.1 | 13.3 | 74.1 | 96.4 | 59.8 | 47.8 | 78.6 | 77.9 | 54.5 | 68.1 | 98.6 | 92.9 | 92.6 | 85.2 | 56.5 | 86.7 | 64.1 | 83.4 | 75.3 | 11 |
| | Gains | +0.8 | +0.0 | -0.2 | -0.4 | +1.0 | +0.5 | -0.3 | -0.2 | +0.6 | -0.8 | +0.8 | -0.7 | +0.2 | +3.4 | +1.2 | -1.5 | -0.6 | +0.4 | +1.1 | +1.7 | +0.4 | |

Table 6: Comparison of CACR and MoCov3 pre-trained ViT-B/16 encoder on ELEVATER benchmark (Li et al., 2022) . We conduct fine-tuning (FT) and linear-probing (LP) in both 5-shot (top 2 rows) and full-show (bottom 2 rows) on 20 datasets. We calculate the gains marked in green for positive results. The mean score and number of wins are reported in the last two columns.

**Downstream image classification:** To measure the efficiency in adapting the pre-trained model to a wide range of downstream data-sets (Kornblith et al., 2021), we employ the recently developed ELEVATER benchmark (Li et al., 2022) to consider both 5-shot and full-shot transfer learning setting: the pre-trained ViT-B/16 is evaluated with fine-tuning and linear probing on 20 public image classification data sets, where for each data set 5 training samples are randomly selected in 5-shot setting, otherwise all data are used to train the model for 50 epochs before the test score is reported, and 3 random seeds are considered for each data set. We deploy the automatic hyper-parameter tuning pipeline implemented in ELEVATER that searches for the best parameter for each model to make a fair fine-tuning and linear probing comparison of pre-trained models. The original metrics of each dataset are used with more details provided in Li et al. (2022) and Appendix C. To measure the overall performance, we consider the average scores over 20 datasets, and "# Wins" indicates the number of data sets on which the current model outperforms its counterpart. As shown in Table 6, we observe in 5-shot scenarios linear probing tends to outperform fine-tuning, likely due to the model being heavily adapted to the pre-training data, thus making it less flexible for new tasks. Despite such as challenge, we can still observe CACR preserves a better transferability with a significant gain. As the amount of task-specific data increases, we observe that fine-tuning starts to outperform linear probing, and CACR still outperforms MoCov3, even though the gap between these two methods become smaller. Overall, in both settings, CACR outperforms MoCov3 in 75% of the downstream datasets, indicating the representation efficiency of transferring in downstream applications.

**Object detection and segmentation:** Besides the linear classification evaluation, following the protocols in previous works (Tian et al., 2019; He et al., 2020; Chen et al., 2020c; Wang & Isola, 2020), we use the

Table 7: Results of transferring features to object detection and segmentation task on Pascal VOC, with the pre-trained ResNet50 on ImageNet-1k. Contrastive learning methods are highlighted.

| Method | | VOC 07+12 detection | | | COCO detection | | | COCO instance seg. | | |
|---|---|---|---|---|---|---|---|---|---|---|
| | | $AP_{50}$ | AP | $AP_{75}$ | $AP_{50}$ | AP | $AP_{75}$ | $AP_{50}$ | AP | $AP_{75}$ |
| scratch | | 60.2 | 33.8 | 33.1 | 44.0 | 26.4 | 27.8 | 46.9 | 29.3 | 30.8 |
| supervised | | 81.3 | 53.5 | 58.8 | 58.2 | 38.2 | 41.2 | 54.7 | 33.3 | 35.2 |
| Non-Contrastive | BYOL | 81.4 | 55.3 | 61.1 | 57.8 | 37.9 | 40.9 | 54.3 | 33.2 | 35.0 |
| | SwAV | 81.5 | 55.4 | 61.4 | 57.6 | 37.6 | 40.3 | 54.2 | 33.1 | 35.1 |
| (wo. Negatives) | SimSiam | 82.4 | 57.0 | 63.7 | 59.3 | 39.2 | 42.1 | **56.0** | 34.4 | 36.7 |
| | Barlow Twins | 82.6 | 56.8 | 63.4 | 59.0 | 39.2 | 42.5 | **56.0** | 34.3 | 36.5 |
| Contrastive | SimCLR | 81.8 | 55.5 | 61.4 | 57.7 | 37.9 | 40.9 | 54.6 | 33.3 | 35.3 |
| | MoCov2 | 82.3 | 57.0 | 63.3 | 58.8 | 39.2 | 42.5 | 55.5 | 34.3 | 36.6 |
| | AU-CL | 82.5 | 57.2 | 63.8 | 58.4 | 39.1 | 42.2 | 55.7 | 34.1 | 36.3 |
| (w. Negatives) | CACR(K=1) | **82.8** | 57.8 | 64.2 | 58.9 | 39.3 | 42.5 | 55.6 | 34.4 | 36.7 |
| | CACR(K=4) | **82.8** | **57.9** | **64.9** | **59.8** | **40.0** | **42.7** | 55.8 | **35.0** | **37.0** |

pretrained ResNet50 on ImageNet-1K for object detection and segmentation task on Pascal VOC (Everingham et al., 2010) and COCO (Lin et al., 2014) by using detectron2 (Wu et al., 2019). The experimental setting details are shown in Appendix C.2 and kept the same as He et al. (2020) and Chen et al. (2020c). The test AP, $AP_{50}$, and $AP_{75}$ of bounding boxes in object detection and test AP, $AP_{50}$, and $AP_{75}$ of masks in segmentation are reported in Table 7. We can observe that the performances of CACR are consistently better than baselines using contrastive objectives, and better than non-contrastive self-supervised learning SOTAs.

## 6    Conclusion

In this paper, we rethink the limitation of conventional contrastive learning (CL) methods that use the contrastive loss but merely consider the intra-relation between samples. In the spirit of a distributional transport between positive and negative samples, we introduce Contrastive Attraction and Contrastive Repulsion (CACR) loss with a doubly contrastive strategy, which constructs for two conditional distributions to respectively model the importance of a positive sample and that of a negative sample to the query according to their distances to the query. Our theoretical analysis and empirical results show that the CACR loss can effectively attract positive samples and repel negative ones from the query as CL intends to do, but is more robust in more general cases. Extensive experiments on small, large-scale, and imbalanced datasets consistently demonstrate the superiority and robustness of CACR over the state-of-the-art methods in contrastive representation learning and related downstream tasks.

**Broader Impact Statement**

Contrastive learning (CL) is effective in learning data representations without label supervision and has led to notable recent progresses in a variety of research areas, such as computer vision. Recently proposed advanced CL methods often require a huge amount of data and thus cost large computational energy. Especially in the case where one needs to use multiple positive pairs in the contrast. Instead of contrasting each positive pair over multiple negative pairs with the classic softmax cross-entropy, our work discovers that the contrastive attraction within positives and contrastive repulsion within negatives bring new insight in self-supervised representation learning. CACR, which naturally takes multiple positive samples in the contrast without making the contrast complexity become combinatorial in the number of positive pairs, has demonstrated clear improvements over existing CL methods. However, the same as existing CL methods, our method is not designed to resist the potential biases existing in the dataset, *e.g.* the false negatives in data. At the current stage, CACR relies on the positive contrast to implicitly alleviate this issue: if a false negative sample is repelled too far away from the query, in the positive attraction, it will be assigned with larger probability to be pulled back. This raises the risk of the quality of learned representations. In the future work, we aim and also encourage other researchers to consider the resistance of these potential risks to make the learned representations more robust and powerful.

**Acknowledgments**

H. Zheng and M. Zhou acknowledge the support of NSF-IIS 2212418 and TACC.

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

## Appendix

## A   Proofs and detailed derivation

***Proof of Property 1.*** By definition, the point-to-point cost $c(\boldsymbol{z}_1, \boldsymbol{z}_2)$ is always non-negative. Without loss of generality, we define it with the Euclidean distance. When equation 6 is true, the expected cost of moving between a pair of positive samples, as defined as $\mathcal{L}_{\text{CA}}$ in equation 2, will reach its minimum at 0. When equation 6 is not true, by definition we will have $\mathcal{L}_{\text{CA}} > 0$, *i.e.*, $\mathcal{L}_{\text{CA}} = 0$ is possible only if equation 6 is true. □

***Proof of Lemma 4.1.*** By changing the reference distribution of the expectation from $\pi_{\boldsymbol{\theta}}^+(\cdot\,|\,\boldsymbol{x}, \boldsymbol{x}_0)$ to $p(\cdot\,|\,\boldsymbol{x}_0)$, we can directly re-write the CA loss as:

$$\mathcal{L}_{\text{CA}} = \mathbb{E}_{\boldsymbol{x} \sim p(\boldsymbol{x})} \mathbb{E}_{\boldsymbol{x}^+ \sim \pi_{\boldsymbol{\theta}}^+(\cdot\,|\,\boldsymbol{x}, \boldsymbol{x}_0)} \left[ c(f_{\boldsymbol{\theta}}(\boldsymbol{x}), f_{\boldsymbol{\theta}}(\boldsymbol{x}^+)) \right]$$

$$= \mathbb{E}_{\boldsymbol{x}_0} \mathbb{E}_{\boldsymbol{x}, \boldsymbol{x}^+ \sim p(\cdot\,|\,\boldsymbol{x}_0)} \left[ -f_{\boldsymbol{\theta}}(\boldsymbol{x})^\top f_{\boldsymbol{\theta}}(\boldsymbol{x}^+) \frac{\pi_{\boldsymbol{\theta}}^+(\boldsymbol{x}^+\,|\,\boldsymbol{x}, \boldsymbol{x}_0)}{p(\boldsymbol{x}^+\,|\,\boldsymbol{x}_0)} \right],$$

which complete the proof. □

***Proof of Lemma 4.2.*** Denoting

$$Z(\boldsymbol{x}) = \int e^{-d(f_{\boldsymbol{\theta}}(\boldsymbol{x}), f_{\boldsymbol{\theta}}(\boldsymbol{x}^-))} p(\boldsymbol{x}^-) d\boldsymbol{x}^-,$$

we have

$$\ln \pi_{\boldsymbol{\theta}}^-(\boldsymbol{x}^-\,|\,\boldsymbol{x}) = -d(f_{\boldsymbol{\theta}}(\boldsymbol{x}), f_{\boldsymbol{\theta}}(\boldsymbol{x}^-)) + \ln p(\boldsymbol{x}^-) - \ln Z(\boldsymbol{x}).$$

Thus we have

$$\mathcal{L}_{\text{CR}} = \mathbb{E}_{\boldsymbol{x} \sim p(\boldsymbol{x})} \mathbb{E}_{\boldsymbol{x}^- \sim \pi_{\boldsymbol{\theta}}^-(\cdot\,|\,\boldsymbol{x})} [\ln \pi_{\boldsymbol{\theta}}^-(\boldsymbol{x}^-\,|\,\boldsymbol{x}) - \ln p(\boldsymbol{x}^-) + \ln Z(\boldsymbol{x})]$$

$$= C_1 + C_2 - \mathcal{H}(X^-\,|\,X) \tag{9}$$

where $C_1 = \mathbb{E}_{\boldsymbol{x} \sim p(\boldsymbol{x})} \mathbb{E}_{\boldsymbol{x}^- \sim \pi_{\boldsymbol{\theta}}^-(\cdot\,|\,\boldsymbol{x})} [\ln p(\boldsymbol{x}^-)]$ and $C_2 = -\mathbb{E}_{\boldsymbol{x} \sim p(\boldsymbol{x})} \ln Z(\boldsymbol{x})$. Under the assumption of a uniform prior on $p(\boldsymbol{x})$, $C_1$ becomes a term that is not related to $\boldsymbol{\theta}$. Under the assumption of a uniform prior on $p(\boldsymbol{z})$, where $\boldsymbol{z} = f_{\boldsymbol{\theta}}(\boldsymbol{x})$, we have

$$Z(\boldsymbol{x}) = \mathbb{E}_{\boldsymbol{x}^- \sim p(\boldsymbol{x})} [e^{-d(f_{\boldsymbol{\theta}}(\boldsymbol{x}), f_{\boldsymbol{\theta}}(\boldsymbol{x}^-))}]$$

$$= \mathbb{E}_{\boldsymbol{z}^- \sim p(\boldsymbol{z})} [e^{-(\boldsymbol{z}^- - \boldsymbol{z})^T (\boldsymbol{z}^- - \boldsymbol{z})}]$$

$$\propto \int e^{-(\boldsymbol{z}^- - \boldsymbol{z})^T (\boldsymbol{z}^- - \boldsymbol{z})} d\boldsymbol{z}^-$$

$$= \sqrt{\pi}, \tag{10}$$

which is also not related to $\boldsymbol{\theta}$. Therefore, under the uniform prior assumption on both $p(\boldsymbol{x})$ and $p(\boldsymbol{z})$, minimizing $\mathcal{L}_{\text{CR}}$ is the same as maximizing $\mathcal{H}(X^-\,|\,X)$, as well as the same as minimizing $I(X, X^-)$. □

***Proof of Lemma 4.3.*** The CL loss can be decomposed as an expected dissimilarity term and a log-sum-exp term:

$$\mathcal{L}_{\text{CL}} := \underset{(\boldsymbol{x}, \boldsymbol{x}^+, \boldsymbol{x}_{1:M}^-)}{\mathbb{E}} \left[ -\ln \frac{e^{f_{\boldsymbol{\theta}}(\boldsymbol{x})^\top f_{\boldsymbol{\theta}}(\boldsymbol{x}^+)/\tau}}{e^{f_{\boldsymbol{\theta}}(\boldsymbol{x})^\top f_{\boldsymbol{\theta}}(\boldsymbol{x}^+)/\tau} + \sum_i e^{f_{\boldsymbol{\theta}}(\boldsymbol{x}_i^-)^\top f_{\boldsymbol{\theta}}(\boldsymbol{x})/\tau}} \right]$$

$$= \mathbb{E}_{(\boldsymbol{x}, \boldsymbol{x}^+)} \left[ -\frac{1}{\tau} f_{\boldsymbol{\theta}}(\boldsymbol{x})^\top f_{\boldsymbol{\theta}}(\boldsymbol{x}^+) \right] + \underset{(\boldsymbol{x}, \boldsymbol{x}^+, \boldsymbol{x}_{1:M}^-)}{\mathbb{E}} \left[ \ln \left( e^{f_{\boldsymbol{\theta}}(\boldsymbol{x})^\top f_{\boldsymbol{\theta}}(\boldsymbol{x}^+)/\tau} + \sum_{i=1}^{M} e^{f_{\boldsymbol{\theta}}(\boldsymbol{x}_i^-)^\top f_{\boldsymbol{\theta}}(\boldsymbol{x})/\tau} \right) \right],$$

where the positive sample $\boldsymbol{x}^+$ is independent of $\boldsymbol{x}$ given $\boldsymbol{x}_0$ and the negative samples $\boldsymbol{x}_i^-$ are independent of $\boldsymbol{x}$. As the number of negative samples goes to infinity, following Wang & Isola (2020), the normalized CL loss is decomposed into the sum of the align loss, which describes the contribution of the positive samples, and the uniform loss, which describes the contribution of the negative samples:

$$\lim_{M\to\infty} \mathcal{L}_{\mathrm{CL}} - \ln M = \underbrace{\mathbb{E}_{(\boldsymbol{x},\boldsymbol{x}^+)}\left[-\frac{1}{\tau} f_{\boldsymbol{\theta}}(\boldsymbol{x})^\top f_{\boldsymbol{\theta}}(\boldsymbol{x}^+)\right]}_{\text{contribution of positive samples}} + \underbrace{\mathbb{E}_{\boldsymbol{x}\sim p(\boldsymbol{x})}\left[\ln \mathbb{E}_{\boldsymbol{x}^-\sim p(\boldsymbol{x}^-)} e^{f_{\boldsymbol{\theta}}(\boldsymbol{x}^-)^\top f_{\boldsymbol{\theta}}(\boldsymbol{x})/\tau}\right]}_{\text{contribution of negative samples}}$$

With importance sampling, the second term in the RHS of the above equation can be further derived into:

$$\mathbb{E}_{\boldsymbol{x}\sim p(\boldsymbol{x})}\left[\ln \mathbb{E}_{\boldsymbol{x}^-\sim p(\boldsymbol{x}^-)} e^{f_{\boldsymbol{\theta}}(\boldsymbol{x}^-)^\top f_{\boldsymbol{\theta}}(\boldsymbol{x})/\tau}\right]$$
$$= \mathbb{E}_{\boldsymbol{x}\sim p(\boldsymbol{x})}\left[\ln \mathbb{E}_{\boldsymbol{x}^-\sim \pi_{\boldsymbol{\theta}}(\boldsymbol{x}^-|\boldsymbol{x})}\left[e^{f_{\boldsymbol{\theta}}(\boldsymbol{x}^-)^\top f_{\boldsymbol{\theta}}(\boldsymbol{x})/\tau} \frac{p(\boldsymbol{x}^-)}{\pi_{\boldsymbol{\theta}}(\boldsymbol{x}^-|\boldsymbol{x})}\right]\right]$$

Apply the Jensen inequality, the second term is decomposed into the negative cost plus a log density ratio:

$$\mathbb{E}_{\boldsymbol{x}\sim p(\boldsymbol{x})}\left[\ln \mathbb{E}_{\boldsymbol{x}^-\sim p(\boldsymbol{x}^-)} e^{f_{\boldsymbol{\theta}}(\boldsymbol{x}^-)^\top f_{\boldsymbol{\theta}}(\boldsymbol{x})/\tau}\right]$$
$$\geqslant \mathbb{E}_{\boldsymbol{x}\sim p(\boldsymbol{x})}\left[\mathbb{E}_{\boldsymbol{x}^-\sim \pi_{\boldsymbol{\theta}}(\boldsymbol{x}^-|\boldsymbol{x})}\left[f_{\boldsymbol{\theta}}(\boldsymbol{x}^-)^\top f_{\boldsymbol{\theta}}(\boldsymbol{x})/\tau\right]\right] + \mathbb{E}_{\boldsymbol{x}\sim p(\boldsymbol{x})}\left[\mathbb{E}_{\boldsymbol{x}^-\sim \pi_{\boldsymbol{\theta}}(\boldsymbol{x}^-|\boldsymbol{x})}\left[\ln \frac{p(\boldsymbol{x}^-)}{\pi_{\boldsymbol{\theta}}(\boldsymbol{x}^-\mid\boldsymbol{x})}\right]\right]$$
$$= \mathbb{E}_{\boldsymbol{x}\sim p(\boldsymbol{x})}\left[\mathbb{E}_{\boldsymbol{x}^-\sim \pi_{\boldsymbol{\theta}}(\boldsymbol{x}^-\mid\boldsymbol{x})}\left[f_{\boldsymbol{\theta}}(\boldsymbol{x}^-)^\top f_{\boldsymbol{\theta}}(\boldsymbol{x})/\tau\right]\right] - I(X; X^-)$$

Defining the point-to-point cost function between two unit-norm vectors as $c(\boldsymbol{z}_1, \boldsymbol{z}_2) = -\boldsymbol{z}_1^\top \boldsymbol{z}_2$ (same as the Euclidean cost since $\|\boldsymbol{z}_1 - \boldsymbol{z}_2\|_2^2/2 = 1 - z_1^\top z_2$ ), we have

$$\mathbb{E}_{\boldsymbol{x}\sim p(\boldsymbol{x})}\left[\ln \mathbb{E}_{\boldsymbol{x}^-\sim p(\boldsymbol{x})} e^{f_{\boldsymbol{\theta}}(\boldsymbol{x}^-)^\top f_{\boldsymbol{\theta}}(\boldsymbol{x})/\tau}\right] + I(X; X^-)$$
$$\geqslant \mathbb{E}_{\boldsymbol{x}\sim p(\boldsymbol{x})}\left[\mathbb{E}_{\boldsymbol{x}^-\sim \pi_{\boldsymbol{\theta}}(\boldsymbol{x}^-\mid\boldsymbol{x})}\left[f_{\boldsymbol{\theta}}(\boldsymbol{x}^-)^\top f_{\boldsymbol{\theta}}(\boldsymbol{x})/\tau\right]\right]$$
$$= -\mathbb{E}_{\boldsymbol{x}\sim p(\boldsymbol{x})}\left[\mathbb{E}_{\boldsymbol{x}^-\sim \pi_{\boldsymbol{\theta}}(\boldsymbol{x}^-\mid\boldsymbol{x})}\left[c(f_{\boldsymbol{\theta}}(\boldsymbol{x}^-), f_{\boldsymbol{\theta}}(\boldsymbol{x}))/\tau\right]\right]$$
$$= \mathcal{L}_{\mathrm{CR}}.$$

This concludes the relation between the contribution of the negative samples in CL and that in CACR.

$\square$

# B  Additional experimental results

In this section, we provide additional results in our experiments, including ablation studies, and corresponding qualitative results.

## B.1  Additional results with AlexNet and ResNet50 encoder on small-scale datasets

Following benchmark works in contrative learning, we add STL-10 dataset to evaluate CACR in small-scale experiments. As an additional results on small-scale datasets, we test the performance of CACR two different encoder backbones. Here we strictly follow the same setting of Wang & Isola (2020) and Robinson et al. (2021), and the results are shown in Table 8 and 9. We can observe with ResNet50 encoder backbone, CACR with single positive or multiple positive pairs consistently outperform the baselines. Compared with the results in Table 8, the CACR shows a more clear improvement over the CL baselines.

Table 8: The top-1 classification accuracy (%) of different contrastive objectives with SimCLR framework on small-scale datasets. All methods follow SimCLR setting and apply AlexNet encoder and trained with 200 epochs.

| Dataset | CL | AU-CL | HN-CL | CACR(K=1) | CMC(K=4) | CACR(K=4) |
|---|---|---|---|---|---|---|
| CIFAR-10 | 83.47 | 83.39 | 83.67 | **83.73** | 85.54 | **86.54** |
| CIFAR-100 | 55.41 | 55.31 | 55.87 | **56.52** | 58.64 | **59.41** |
| STL-10 | 83.89 | 84.43 | 83.27 | **84.51** | 84.50 | **85.59** |

Table 9: The top-1 classification accuracy (%) of different contrastive objectives with SimCLR framework on small-scale datasets. All methods follow SimCLR setting and apply a ResNet50 encoder and trained with 400 epochs.

| Dataset | CL | AU-CL | HN-CL | CACR(K=1) | CMC(K=4) | CACR(K=4) |
|---|---|---|---|---|---|---|
| CIFAR-10 | 88.70 | 88.63 | 89.02 | **90.97** | 90.05 | **92.89** |
| CIFAR-100 | 62.00 | 62.57 | 62.96 | **62.98** | 65.19 | **66.52** |
| STL-10 | 84.60 | 83.81 | 84.29 | **88.42** | 91.40 | **93.04** |

## B.2  On the effects of conditional distribution

**Supplementary studies of CA and CR:** As a continuous ablation study shown in Figure 3, we also conduct similar experiments on CIFAR-100, where we study the evolution of conditional entropy $\mathcal{H}(X^-|X)$ *w.r.t.* the training epoch. The results are shown in Figure 4, and the results of exponential label-imbalanced data are shown in Figure 5. Similar to the observation on CIFAR-10, shown in Figure 3, we can observe $\mathcal{H}(X^-|X)$ is getting maximized as the encoder is getting optimized with these methods, as suggested in Lemma 4.2. In the right panel, We can observe baseline methods have lower conditional entropy, which indicates the encoder is less effective in distinguish the nagative samples from query, while CACR consistently provides better performance than the other methods indicating the better robustness of CACR.

As a qualitative verification, we randomly take a query from a mini-batch, and illustrate its positive and negative samples and their conditional probabilities in Figure 6. As shown, given this query of a dog image, the positive sample with the largest weight contains partial dog information, indicating the encoder to focus on texture information; the negatives with larger weights are more related to the dog category, which encourages the encoder to focus on distinguishing these "hard" negative samples. In total, the weights learned by CACR enjoy the interpretability compared to the conventional CL.

We study different definition of the conditional distribution. From Table 10, we can observe that the results are not sensitive to the distance space. In addition, as we change $\pi_+$ to assign larger probability to closer samples, the results are similar to those using single positive pair (K=1). Moreover, the performance drops if we change $\pi_-$ to assign larger probability to more distant negative samples.

**Uniform Attraction and Uniform Repulsion: A degenerated version of CACR**

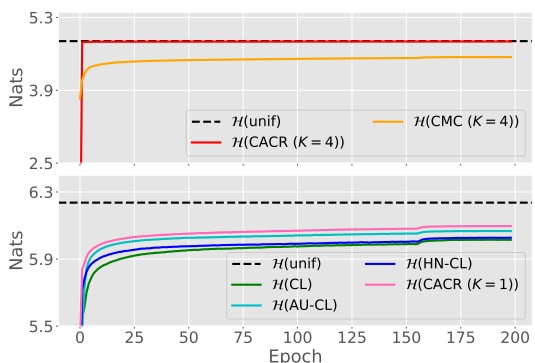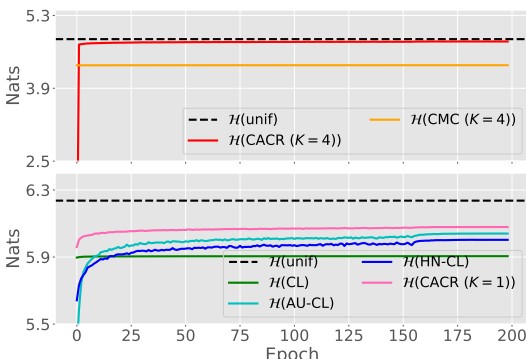

Figure 4: (*Supplementary to Figure 3*) Conditional entropy $\mathcal{H}(X^-|X)$ *w.r.t.* training epoch on CIFAR-100 (**left**) and linear label-imbalanced CIFAR-100 (**right**). The maximal possible conditional entropy is indicated by a dotted line.

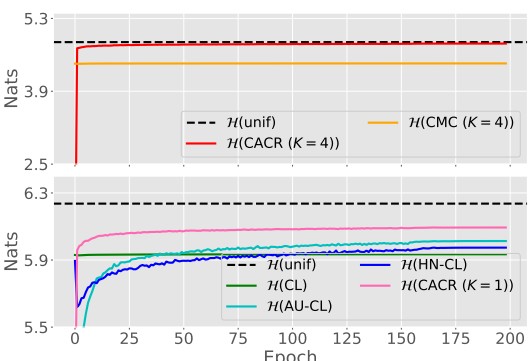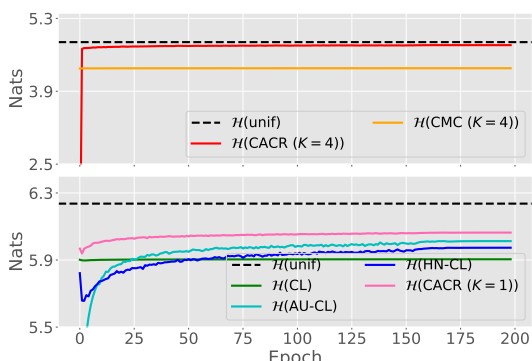

Figure 5: (*Supplementary to Figure 3*) Conditional entropy $\mathcal{H}(X^-|X)$ *w.r.t.* training epoch on exponential label-imbalanced CIFAR-10 (**left**) and CIFAR-100 (**right**). The maximal possible conditional entropy is indicated by a dotted line.

To reinforce the necessity of the contrasts within positives and negatives before the attraction and repulsion, we introduce a degenerated version of CACR here, where the conditional distributions are forced to be uniform. Remind $c(\boldsymbol{z}_1, \boldsymbol{z}_2)$ as the point-to-point cost of moving between two vectors $\boldsymbol{z}_1$ and $\boldsymbol{z}_2$, *e.g.*, the squared Euclidean distance $\|\boldsymbol{z}_1 - \boldsymbol{z}_2\|^2$ or the negative inner product $-\boldsymbol{z}_1^T\boldsymbol{z}_2$. In the same spirit of equation 1, we have considered a uniform attraction and uniform repulsion (UAUR) without doubly contrasts within positive and negative samples, whose objective is

$$\min_{\boldsymbol{\theta}} \left\{ \mathbb{E}_{\boldsymbol{x}_0 \sim p_{data}(\boldsymbol{x})} \mathbb{E}_{\epsilon_0, \epsilon^+ \sim p(\epsilon)} \left[ c(f_{\boldsymbol{\theta}}(\boldsymbol{x}), f_{\boldsymbol{\theta}}(\boldsymbol{x}^+)) \right] - \mathbb{E}_{\boldsymbol{x}, \boldsymbol{x}^- \sim p(\boldsymbol{x})} \left[ c(f_{\boldsymbol{\theta}}(\boldsymbol{x}), f_{\boldsymbol{\theta}}(\boldsymbol{x}^-)) \right] \right\}. \tag{11}$$

The intuition of UAUR is to minimize/maximize the expected cost of moving the representations of positive/negative samples to that of the query, with the costs of all sample pairs being uniformly weighted. While equation 1 has been proven to be effective for representation learning, our experimental results do not find equation 11 to perform well, suggesting that the success of representation learning is not guaranteed by uniformly pulling positive samples towards and pushing negative samples away from the query.

**Distinction between CACR and UAUR**: Compared to UAUR in equation 11 that uniformly weighs different pairs, CACR is distinct in considering the dependency between samples: as the latent-space distance between the query and its positive sample becomes larger, the conditional probability becomes higher, encouraging the encoder to focus more on the alignment of this pair. In the opposite, as the distance between the query and its negative sample becomes smaller, the conditional probability becomes higher, encouraging the encoder to push them away from each other.

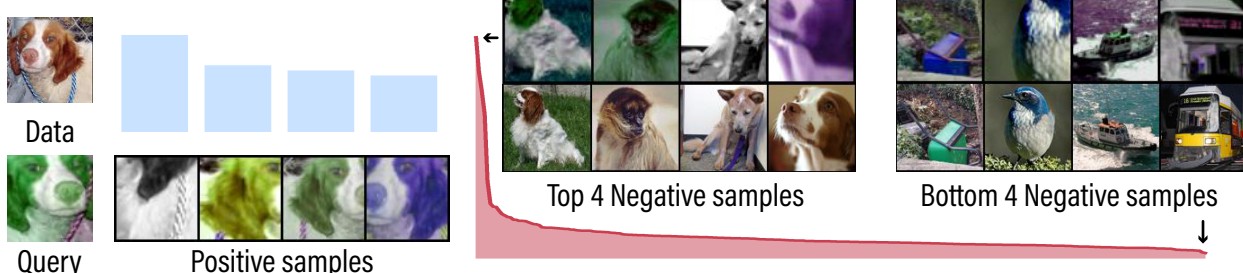

Data

Query    Positive samples    Top 4 Negative samples    Bottom 4 Negative samples

Figure 6: Illustration of positive/negative samples and their corresponding weights. (*Left*) For a query augmented from the original dog image, 4 positive samples are shown, with their weights visualized as the blue distribution. (*Right*) The sampling weights for negative samples are visualized as the red distribution; we visualize 4 negative samples with the highest and 4 with the lowest weights, with their original images shown below.

Table 10: Linear classification performance (%) of different variants of conditional probability. This experiment is done on CIFAR-10, with $K = 4$ and mini-batch size $M = 128$.

| | $\pi_+$ | |
| --- | --- | --- |
| | $\dfrac{e^{+d_{t^+}(f_{\boldsymbol{\theta}}(\boldsymbol{x}), f_{\boldsymbol{\theta}}(\boldsymbol{x}^+))}p(\boldsymbol{x}^+\mid\boldsymbol{x}_0)}{\int e^{+d_{t^+}(f_{\boldsymbol{\theta}}(\boldsymbol{x}), f_{\boldsymbol{\theta}}(\boldsymbol{x}^+))}p(\boldsymbol{x}^+\mid\boldsymbol{x}_0)d\boldsymbol{x}^+}$ | $\dfrac{e^{-d_{t^+}(f_{\boldsymbol{\theta}}(\boldsymbol{x}), f_{\boldsymbol{\theta}}(\boldsymbol{x}^+))}p(\boldsymbol{x}^+\mid\boldsymbol{x}_0)}{\int e^{-d_{t^+}(f_{\boldsymbol{\theta}}(\boldsymbol{x}), f_{\boldsymbol{\theta}}(\boldsymbol{x}^+))}p(\boldsymbol{x}^+\mid\boldsymbol{x}_0)d\boldsymbol{x}^+}$ |
| $\pi_-$ $\dfrac{e^{-d_{t^-}(f_{\boldsymbol{\theta}}(\boldsymbol{x}), f_{\boldsymbol{\theta}}(\boldsymbol{x}^-))}p(\boldsymbol{x}^-)}{\int e^{-d_{t^-}(f_{\boldsymbol{\theta}}(\boldsymbol{x}), f_{\boldsymbol{\theta}}(\boldsymbol{x}^-))}p(\boldsymbol{x}^-)d\boldsymbol{x}^-}$ | 86.48 | 83.91 |
| $\dfrac{e^{+d_{t^-}(f_{\boldsymbol{\theta}}(\boldsymbol{x}), f_{\boldsymbol{\theta}}(\boldsymbol{x}^-))}p(\boldsymbol{x}^-)}{\int e^{+d_{t^-}(f_{\boldsymbol{\theta}}(\boldsymbol{x}), f_{\boldsymbol{\theta}}(\boldsymbol{x}^-))}p(\boldsymbol{x}^-)d\boldsymbol{x}^-}$ | 79.46 | 74.91 |

In order to further explore the effects of the conditional distribution, we conduct an ablation study to compare the performance of different variants of CACR with/without conditional distributions. Here, we compare 4 configurations of CACR ($K = 4$): (*i*) CACR with both positive and negative conditional distribution; (*ii*) CACR without the positive conditional distribution; (*iii*) CACR without the negative conditional distribution; (*iv*) CACR without both positive and negative conditional distributions, which refers to UAUR model (see Equation 11). As shown in Table 11, when discarding the positive conditional distribution, the linear classification accuracy slightly drops. As the negative conditional distribution is discarded, there is a large performance drop compared to the full CACR objective. With the modeling of neither positive nor negative conditional distribution, the UAUR shows a continuous performance drop, suggesting that the success of representation learning is not guaranteed by uniformly pulling positive samples closer and pushing negative samples away. The comparison between these CACR variants shows the necessity of the conditional distribution.

Table 11: Linear classification performance (%) of different variants of our method. "CACR" represents the normal CACR configuration, "w/o $\pi_{\boldsymbol{\theta}}^+$" means without the positive conditional distribution, "w/o $\pi_{\boldsymbol{\theta}}^-$" means without the negative conditional distribution. "UAUR" indicates the uniform cost (see the model we discussed in Equation 11), *i.e.* without both positive and negative conditional distribution. This experiment is done on all small-scale datasets, with $K = 4$ and mini-batch size $M = 128$.

| Methods | CIFAR-10 | CIFAR-100 | STL-10 |
| --- | --- | --- | --- |
| CACR | **85.94** | **59.51** | **85.59** |
| w/o $\pi_{\boldsymbol{\theta}}^+$ | 85.22 | 58.74 | 85.06 |
| w/o $\pi_{\boldsymbol{\theta}}^-$ | 78.49 | 47.88 | 72.94 |
| UAUR | 77.17 | 44.24 | 71.88 |

As qualitative illustrations, we randomly fix one mini-batch, and randomly select one sample as the query. Then we extract the features with the encoder trained with CL loss and CACR ($K = 1$) loss at epochs 1, 20, and 200, and visualize the (four) positives and negatives in the embedding space with $t$-SNE (van der Maaten & Hinton, 2008). For more clear illustration, we center the query in the middle of the plot and only show samples appearing in the range of $[-10, 10]$ on both $x$ and $y$ axis. The results are shown in Figure 7(c), from which we can find that as the the encoder is getting trained, the positive samples are aligned closer and the negative samples are pushed away for both methods. Compared to the encoder trained with CL, we can observe CACR shows better performance in achieving this goal. Moreover, we can observe the distance between any two data points in the plot is more uniform, which confirms that CACR shows better results in the maximization of the conditional entropy $\mathcal{H}(X^-|X)$.

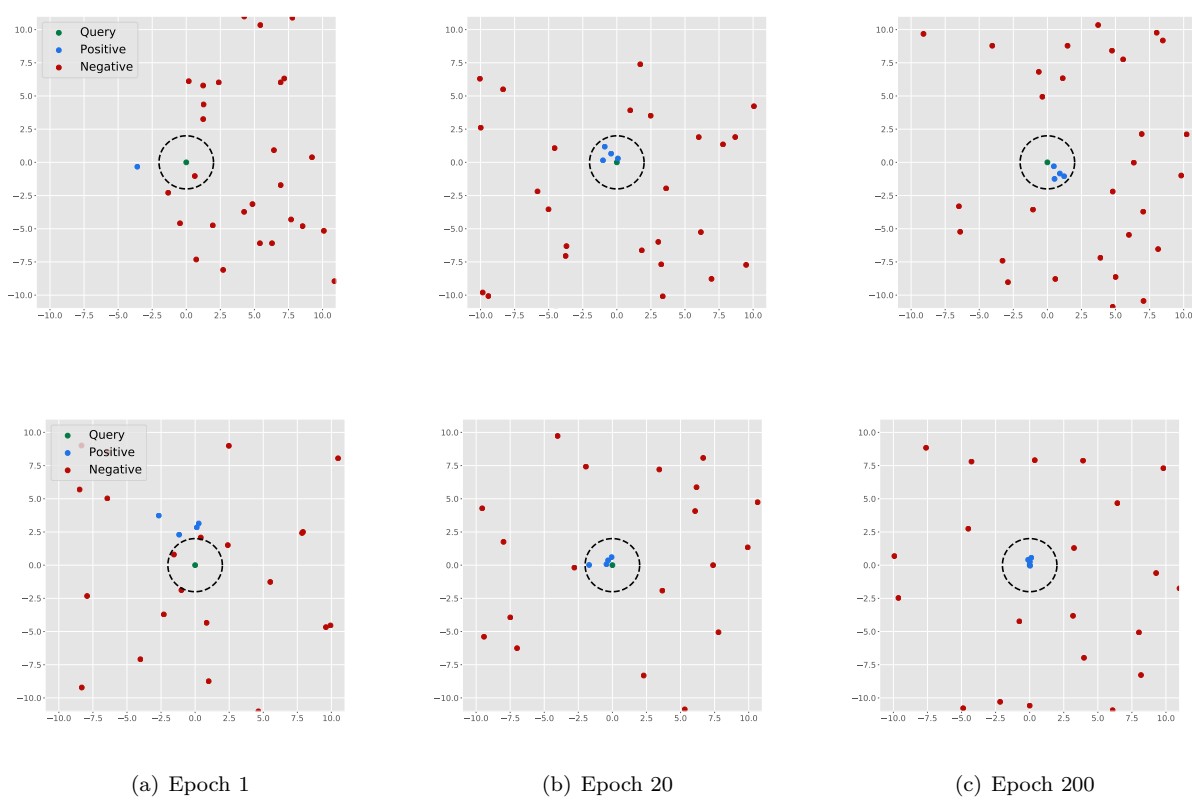

(a) Epoch 1          (b) Epoch 20          (c) Epoch 200

Figure 7: The $t$-SNE visualization of the latent space at different training epochs, learned by CL loss (*top*) and CACR loss (*bottom*). The picked query is marked in green, with its positive samples marked in blue and its negative samples marked in red. The circle with radius $t^-$ is shown as the black dashed line. As the encoder gets trained, we can observe the positive samples are aligned closer to the query (Property 1), and the conditional differential entropy $\mathcal{H}(X^-|X)$ is progressively maximized, driving the distances $d(f_{\boldsymbol{\theta}}(\boldsymbol{x}), f_{\boldsymbol{\theta}}(\boldsymbol{x}^-))$ towards uniform (Lemma 4.2).

### B.3 Ablation study

**On the effects of negative sampling size:** We investigate the model performance and robustness with different sampling size by varying the mini-batch size used in the training. On all the small-scale datasets, the mini-batches are applied with size 64, 128, 256, 512, 768 and the corresponding linear classification results are shown in Figure 8. From this figure, we can see that CACR ($K = 4$) consistently achieves better performance than other objectives. For example, when mini-batch size is 256, CACR ($K = 4$) outperforms CMC by about 0.4%-1.2%. CACR ($K = 1$) shows better performance in most of the cases, while slightly underperforms

than the baselines with mini-batch size 64. A possible explanation could be the estimation of the conditional distribution needs more samples to provide good guidance for the encoder.

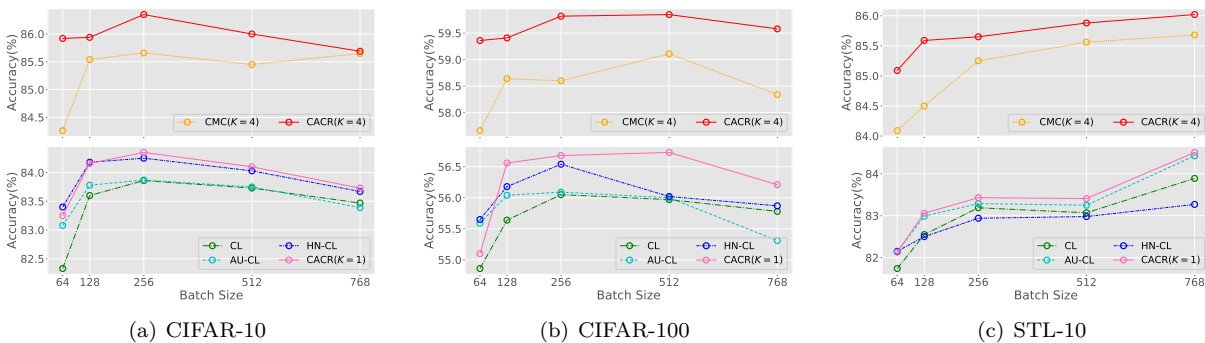

(a) CIFAR-10          (b) CIFAR-100          (c) STL-10

Figure 8: The linear classification results of training with different sampling size on small-scale datasets. The training batch size is proportional to the negative sampling size.

**On the effects of positive sampling size:** We conduct experiments to investigate the model performance with different positive sampling size by using different $K$ values in the pretraining: $K \in \{1, 2, 4, 6, 8, 10\}$ on CIFAR-10/100 and $K \in \{1, 2, 3, 4\}$ on ImageNet-1K. Similar to our experiment setting, in 200 epochs, we apply AlexNet encoder on CIFAR-10 and CIFAR-100 and apply ResNet50 encoder on ImageNet-1K. Shown in Figure 9, we can observe as $K$ increases, the linear classification accuracy increases accordingly.

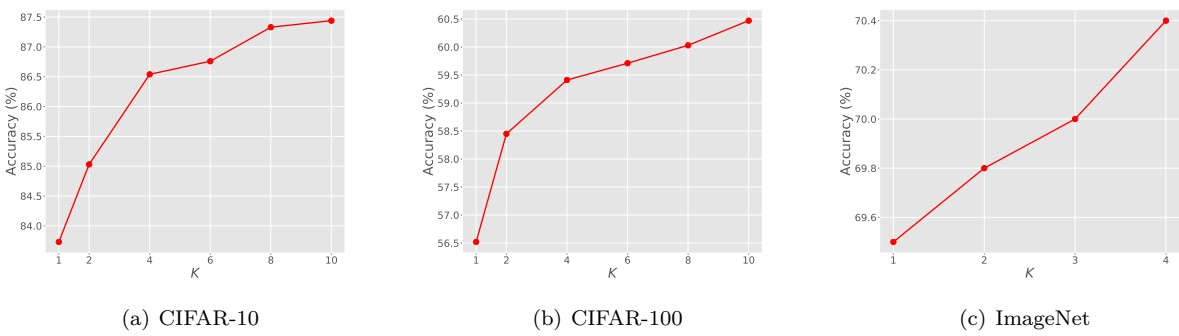

(a) CIFAR-10          (b) CIFAR-100          (c) ImageNet

Figure 9: The linear classification results of training with different positive sampling size on CIFAR-10, CIFAR-100 and ImageNet-1K. An AlexNet encoder is applied on CIFAR-10 and CIFAR-100; ResNet50 encoder is applied on ImageNet.

**On the effects of hyper-parameter $t^+$, $t^-$:** Remind in the definition of positive and negative conditional distribution, two hyper-parameters $t^+$ and $t^-$ are involved as following:

$$\pi_{\boldsymbol{\theta}}^+(\boldsymbol{x}^+ \mid \boldsymbol{x}, \boldsymbol{x}_0) := \frac{e^{t^+ \|f_{\boldsymbol{\theta}}(\boldsymbol{x}) - f_{\boldsymbol{\theta}}(\boldsymbol{x}^+)\|^2} p(\boldsymbol{x}^+ \mid \boldsymbol{x}_0)}{\int e^{t^+ \|f_{\boldsymbol{\theta}}(\boldsymbol{x}) - f_{\boldsymbol{\theta}}(\boldsymbol{x}^+)\|^2} p(\boldsymbol{x}^+ \mid \boldsymbol{x}_0) d\boldsymbol{x}^+}; \quad \pi_{\boldsymbol{\theta}}^-(\boldsymbol{x}^- \mid \boldsymbol{x}) := \frac{e^{-t^- \|f_{\boldsymbol{\theta}}(\boldsymbol{x}) - f_{\boldsymbol{\theta}}(\boldsymbol{x}^-)\|^2} p(\boldsymbol{x}^-)}{\int e^{-t^- \|f_{\boldsymbol{\theta}}(\boldsymbol{x}) - f_{\boldsymbol{\theta}}(\boldsymbol{x}^-)\|^2} p(\boldsymbol{x}^-) d\boldsymbol{x}^-}.$$

In this part, we investigate the effects of $t^+$ and $t^-$ on representation learning performance on small-scale datasets, with mini-batch size 768 ($K = 1$) and 128 ($K = 4$) as an ablation study. We search in a range $\{0.5, 0.7, 0.9, 1.0, 2.0, 3.0\}$. The results are shown in Table 12 and Table 13.

As shown in these two tables, from Table 12, we observe the CACR shows better performance with smaller values for $t^+$. Especially when $t^+$ increases to 3.0, the performance drops up to about 1.9% on CIFAR-100. For analysis, since we have $K = 4$ positive samples for the computation of positive conditional distribution,

Table 12: The classification accuracy(%) of CACR ($K = 4$, $M = 128$) with different hyper-parameters $t^+$ on small-scale datasets.

| Method | Dataset | 0.5 | 0.7 | 0.9 | 1.0 | 2.0 | 3.0 |
|---|---|---|---|---|---|---|---|
| | CIFAR-10 | 86.07 | 85.78 | 85.90 | **86.54** | 84.85 | 84.76 |
| CACR ($K = 4$) | CIFAR-100 | **59.47** | 59.61 | 59.41 | 59.41 | 57.82 | 57.55 |
| | STL-10 | 85.90 | **85.91** | 85.81 | 85.59 | 85.65 | 85.14 |

using a large value for $t^+$ could result in an over-sparse conditional distribution, where the conditional probability is dominant by one or two positive samples. This also explains why the performance when $t^+ = 3.0$ is close to the classification accuracy of CACR ($K = 1$).

Similarly, from Table 13, we can see that a small value for $t^-$ will lead to the degenerated performance. Here, since we are using mini-batches of size 768 ($K = 1$) and 128 ($K = 4$), a small value for $t^-$ will flatten the weights of the negative pairs and make the conditional distribution closer to a uniform distribution, which explains why the performance when $t^- = 0.5$ is close to those without modeling $\pi_{\boldsymbol{\theta}}^-$. Based on these results, the values of $t^+ \in [0.5, 1.0]$ and $t^- \in [0.9, 2.0]$ could be good empirical choices according to our experiment settings on these datasets.

### B.4 Additional comparisons

In this part we provide more comparisons with baseline methods. For small-scale experiments, we still compare with contrastive learning methods, conventional CL loss, align-uniform loss, and hard negative sampling CL loss. For large-scale experiments, we continue to compare with contrastive learning loss on ImageNet-100 and ImagNet-1K with MoCov2 framework, and provide comparisons with SOTAs pretrained with different epochs.

**Training efficiency on small-scale datasets:** On CIFAR-10, CIFAR-100 and STL-10, we pretrained AlexNet encoder in 200 epochs and save linear classification results with learned representations every 10 epochs. Shown in Figure 10, CACR consistently outperforms the other methods in linear classification with the learned representations at the same epoch, indicating a superior learning efficiency of CACR. Correspondingly, we also evaluate the GPU time of CACR loss with different choices of K, as shown in Table 14.

**Comparison with contrastive learning methods on ImageNet:** For large-scale experiments, following the convention, we adapt all methods into the MoCo-v2 framework and pre-train a ResNet50 encoder in 200 epochs with mini-batch size 128/256 on ImageNet-100/ImageNet-1k. Table 15 summarizes the results of linear classification on these two large-scale datasets. Similar to the case on small-scale datasets, CACR consistently shows better performance, improving the baselines at least by 1.74% on ImageNet-100 and 0.71% on ImageNet-1K. In MoCo-v2, with multiple positive samples, CACR improves the baseline methods by 2.92% on ImageNet-100 and 2.75% on ImageNet-1K. It is worth highlighting that the improvement of CACR is more significant on these large-scale datasets, where the data distribution could be much more diverse compared to these small-scale ones. This is not surprising, as according to our theoretical analysis, CACR's

Table 13: The classification accuracy(%) of CACR ($K = 1$, $M = 768$) and CACR ($K = 4$, $M = 128$) with different hyper-parameters $t^-$ on small-scale datasets.

| Methods | Dataset | 0.5 | 0.7 | 0.9 | 1.0 | 2.0 | 3.0 |
|---|---|---|---|---|---|---|---|
| | CIFAR-10 | 81.66 | 82.40 | 83.07 | 82.74 | **83.73** | 83.11 |
| CACR ($K = 1$) | CIFAR-100 | 51.42 | 52.81 | 53.36 | 54.20 | 56.21 | **56.52** |
| | STL-10 | 80.37 | 81.47 | 84.46 | 82.16 | 84.21 | **84.51** |
| | CIFAR-10 | 85.67 | 86.19 | **86.54** | 86.41 | 85.94 | 85.69 |
| CACR ($K = 4$) | CIFAR-100 | 58.17 | 58.63 | 59.37 | 59.35 | **59.41** | 59.31 |
| | STL-10 | 83.81 | 84.42 | 84.71 | 85.25 | **85.59** | 85.41 |

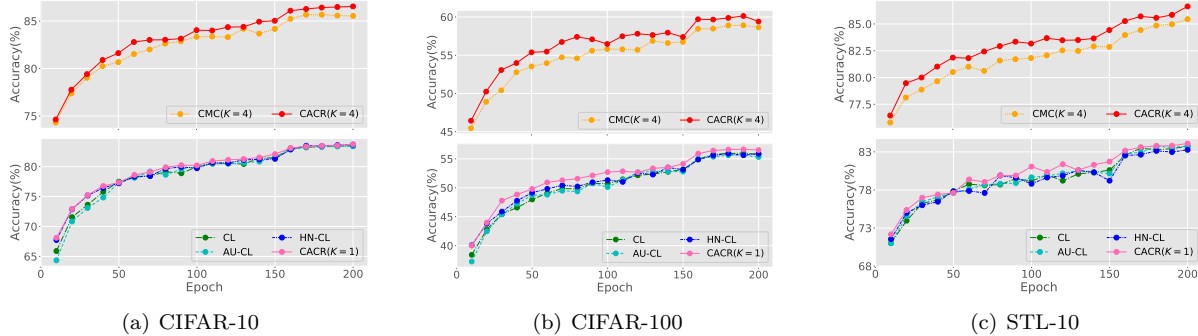

Figure 10: Comparison of training efficientcy: Linear classification with learned representations *w.r.t.* training epoch on CIFAR-10, CIFAR-100 and STL-10.

Table 14: GPU time (s) per iteration of CACR *w.r.t.* different $K$ on CIFAR-10 with AlexNet framework (mini-batch size is 128), tested on Tesla-v100 GPU.

| K | 1 | 2 | 4 | 6 | 8 | 10 |
|---|---|---|---|---|---|---|
| GPU time (s) / iteration | 0.0021 | 0.0026 | 0.0035 | 0.0045 | 0.0054 | 0.0064 |

double-contrast within samples enhances the effectiveness of the encoder's optimization. Moreover, we can see CACR ($K = 1$) shows a clear improvement over HN-CL. A possible explanation is that although both increasing the negative sample size and selecting hard negatives are proposed to improve the CL loss, the effectiveness of hard negatives is limited when the sampling size is increased over a certain limit. As CACR targets to repel the negative samples away, the conditional distribution still efficiently guides the repulsion when the sampling size becomes large.

Table 15: Comparison with contrastive learning methods: Top-1 classification accuracy (%) of different contrastive learning objectives on MoCo-v2 framework and ResNet50 encoder, pretrained on ImageNet-1K dataset with 200 epochs. The results from paper or Github page are marked by $\star$.

| Methods | ImageNet-100 | ImageNet-1K |
|---|---|---|
| MoCov2 (CL) | 77.54$^\star$ | 67.50$^\star$ |
| AU-CL | 77.66$^\star$ | 67.69$^\star$ |
| HN-CL | 76.34 | 67.41 |
| CACR ($K = 1$) | **79.40** | **68.40** |
| CMC (CL, $K = 4$) | 78.84 | 69.45 |
| CACR ($K = 4$) | **80.46** | **70.35** |

Table 16: Comparison with state-of-the-arts on linear probe classification accuracy, pretrained with different epochs, using ResNet50 encoder backbone on ImageNet-1k.

| Epochs | 100 | 200 | 400 | 800 | 1000 |
|---|---|---|---|---|---|
| BYOL | 66.5 | 70.6 | 73.2 | **74.3** | - |
| BarlowTwins | - | - | - | - | 73.2 |
| SWAV | 66.5 | 69.1 | 70.7 | 71.8 | - |
| Simsiam | 68.1 | 70.0 | 70.8 | 71.3 | - |
| SimCLR | 66.5 | 68.3 | 69.8 | 70.4 | 71.7 |
| MoCov2 | 67.4 | 69.9 | 71.0 | 72.2 | - |
| FNC (multi-crop) | 70.4 | - | - | - | 74.4 |
| CACR | 68.3 | **70.4** | **73.8** | 74.0 | **74.4** |

**Comparison with other SOTAs:** Besides the methods using contrastive loss, we continue to compare with the self-supervised learning methods like BYOL, SWaV, SimSiam, **etc.** that do not involve the contrasts with negative samples. Table 16 provides more detailed comparison with all state-of-the-arts in different epochs and could better support the effectiveness of CACR: We can observe CACR achieves competitive results and generally outperforms most of SOTAs at the same epoch in linear classification tasks. We also compare the computation complexity. Table 17 reports computation complexity to provide quantitative results in terms of positive number K, where we can observe the computation cost of CACR slightly increases as K increase, but does not increase as that when using multi-positives in CL loss.

Table 17: GPU time (s) per iteration of different loss on MoCov2 framework, tested on 32G-V100 GPU

| Methods | CL | AU-CL | HN-CL | CACR(K=1) | CL (K=4) | CACR(K=2) | CACR(K=3) | CACR(K=4) |
|---|---|---|---|---|---|---|---|---|
| Batch size M | 256 | 256 | 256 | 256 | 64 | 128 | 64 | 64 |
| # samples (KxM) / iteration | 256 | 256 | 256 | 256 | 256 | 256 | 192 | 256 |
| GPU time (s) / iteration | 0.837 | 0.840 | 0.889 | 0.871 | 3.550 | 0.996 | 1.017 | 1.342 |

| Method | ResNet50 | | ViT-B/16 | |
|---|---|---|---|---|
| | FT | Lin-cls | FT | Lin-cls |
| SimCLRv2 | 77.2 | 71.7 | 83.1 | 73.9 |
| MoCov3 | 77.0 | 73.8 | 83.2 | 76.5 |
| CACR | 78.1 | **74.7** | **83.4** | **76.8** |
| SWAV$^\dagger$ | 77.8 | 75.3 | 82.8 | 71.6 |
| CACR$^\dagger$ | **78.4** | 75.3 | **83.4** | **77.1** |

Table 18: Comparison with state-of-the-arts on fine-tuning and linear probing classification accuracy (%), pre-trained using ResNet50 and ViT-Base/16 encoder backbone on ImageNet-1k. $^\dagger$ indicates using SWAV multi-crops.

**Comparison with advanced architectures:** Beyond the conventional evaluation on linear probing, recent self-supervised learning methods use advanced encoder architecture such as Vision Transformers (ViT) (Dosovitskiy et al., 2020), and are evaluated with end-to-end fine-tuning. We incorporate these perspectives with CACR for a complete comparison. Table 18 provides a comparison with the state-of-the-arts using ResNet50 and ViT-Base/16 as backbone, where we follow their experiment settings and pre-train ResNet50 with 800 epochs and ViT-B/16 with 300 epochs. We can observe CACR generally outperforms these methods in both fine-tuning and linear probing classification tasks.

| CLIP Radford et al. (2021) | CLIP-reproduced | CACR |
|---|---|---|
| 19.8 | 19.2 | **22.7** |

Table 19: Top-1 zero-shot classification accuracy (%) on ImageNet1K, pre-trained using ResNet50 on CC3M dataset.

**Multi-modal contrastive learning:** Besides self-supervised learning on vision tasks, we follow CLIP Radford et al. (2021) to evaluate CACR on multi-modal representation learning. We compare CACR's performance with CLIP, with our reproduced result and the results reported in Li et al. (2022) in Table 19. All methods are pre-trained on CC3M dataset with ResNet50 backbone for 32 epochs. We can observe CACR surpasses CLIP by 2.9% in terms of zero-shot accuracy on ImageNet.

### B.5 Connection to other representation learning methods

**Results of different cost metrics**

Recall that the definition of the point-to-point cost metric is usually set as the quadratic Euclidean distance:

$$c(f_\theta(\boldsymbol{x}), f_\theta(\boldsymbol{y})) = ||f_\theta(\boldsymbol{x}) - f_\theta(\boldsymbol{y})||_2^2. \tag{12}$$

In practice, the cost metric defined in our method is flexible to be any valid metrics. Here, we also investigate the performance when using the Radial Basis Function (RBF) cost metrics:

$$c_{\mathrm{RBF}}(f_\theta(\boldsymbol{x}), f_\theta(\boldsymbol{y})) = -e^{-t||f_\theta(\boldsymbol{x}) - f_\theta(\boldsymbol{y})||_2^2}, \tag{13}$$

where $t \in \mathbb{R}^+$ is the precision of the Gaussian kernel. With this definition of the cost metric, our method is closely related to the baseline method AU-CL (Wang & Isola, 2020), where the authors calculate pair-wise RBF cost for the loss *w.r.t.* negative samples. Following Wang & Isola (2020), we replace the cost metric

when calculate the negative repulsion cost with the RBF cost and modify $\hat{\mathcal{L}}_{\text{CR}}$ as:

$$\hat{\mathcal{L}}_{\text{CR}-\text{RBF}} := \ln\Big[\frac{1}{M}\sum_{i=1}^{M}\sum_{j\neq i}\frac{e^{-d_{t^-}(f_{\boldsymbol{\theta}}(\boldsymbol{x}_i),f_{\boldsymbol{\theta}}(\boldsymbol{x}_j))}}{\sum_{j'\neq i}e^{-d_{t^-}(f_{\boldsymbol{\theta}}(\boldsymbol{x}_i),f_{\boldsymbol{\theta}}(\boldsymbol{x}_{j'}))}}\times c_{\text{RBF}}(f_{\boldsymbol{\theta}}(\boldsymbol{x}_i),f_{\boldsymbol{\theta}}(\boldsymbol{x}_j))\Big] \tag{14}$$

$$= \ln\Big[\frac{1}{M}\sum_{i=1}^{M}\sum_{j\neq i}\frac{e^{-d_{t^-}(f_{\boldsymbol{\theta}}(\boldsymbol{x}_i),f_{\boldsymbol{\theta}}(\boldsymbol{x}_j))}}{\sum_{j'\neq i}e^{-d_{t^-}(f_{\boldsymbol{\theta}}(\boldsymbol{x}_i),f_{\boldsymbol{\theta}}(\boldsymbol{x}_{j'}))}}\times e^{-t\|f_{\boldsymbol{\theta}}(\boldsymbol{x}_i)-f_{\boldsymbol{\theta}}(\boldsymbol{x}_j)\|^2}\Big].$$

Here the negative cost is in log scale for numerical stability. When using the RBF cost metric, we use the same setting in the previous experiments and evaluate the linear classification on all small-scale datasets. The results of using Euclidean and RBF cost metrics are shown in Table 20. From this table, we see that both metrics achieve comparable performance, suggesting the RBF cost is also valid in our framework. In CACR, the cost metric measures the cost of different sample pairs and is not limited on specific formulations. More favorable cost metrics can be explored in the future.

Table 20: The classification accuracy (%) of CACR ($K = 1$) and CACR ($K = 4$) with different cost metrics on CIFAR-10, CIFAR-100 and STL-10. Euclidean indicates the cost defined in 12, and RBF indicates the cost metrics defined in 13.

| Methods | Cost Metric | CIFAR-10 | CIFAR-100 | STL-10 |
|---|---|---|---|---|
| CACR($K = 1$) | Euclidean | 83.73 | 56.21 | 83.55 |
| | RBF | 83.08 | 55.90 | 84.20 |
| CACR($K = 4$) | Euclidean | 85.94 | **59.41** | 85.59 |
| | RBF | **86.20** | 58.81 | **85.80** |

**Discussion: Relation to triplet loss** CACR is also related to the widely used triplet loss (Schroff et al., 2015; Sun et al., 2020b). A degenerated version of CACR where the conditional distributions are all uniform can be viewed as triplet loss, while underperform the proposed CACR, as discussed in Section B.2. In the view of triplet loss, CACR is dealing with the margin between expected positive pair similarity and negative similarity:

$$\mathcal{L}_{\text{CACR}} = [\mathbb{E}_{\pi_{t^+}(\boldsymbol{x}^+|x)}[c(\boldsymbol{x},\boldsymbol{x}^+)] - \mathbb{E}_{\pi_{t^-}(\boldsymbol{x}^-|\boldsymbol{x})}[c(\boldsymbol{x},\boldsymbol{x}^-)] + m]_+$$

which degenerates to the generic triplet loss if the conditional distribution degenerates to a uniform distribution:

$$\mathcal{L}_{\text{UAUR}} = [\mathbb{E}_{p(\boldsymbol{x}^+)}[c(\boldsymbol{x},\boldsymbol{x}^+)] - \mathbb{E}_{p(\boldsymbol{x}^-)}[c(\boldsymbol{x},\boldsymbol{x}^-)] + m]_+ = [c(\boldsymbol{x},\boldsymbol{x}^+) - c(\boldsymbol{x},\boldsymbol{x}^-) + m]_+$$

This degeneration also highlights the importance of the Bayesian derivation of the conditional distribution. The experimental results of the comparison between CACR and the degenerated uniform version (equivalent to generic triplet loss) are presented in Table 11.

Moreover, CACR loss can degenerate to a triplet loss with hard example mining if $\pi_{t^+}(x^+|x)$ and $\pi_{t^+}(x^+|x)$ are sufficiently concentrated, where the density shows a very sharp peak:

$$\mathcal{L}_{\text{CACR}} = [\max(c(\boldsymbol{x},\boldsymbol{x}^+)) - \min(c(\boldsymbol{x},\boldsymbol{x}^-)) + m]_+$$

which corresponds to the loss shown in Schroff et al. (2015). As shown in Table 12 and 13, when varying $t^+$ and $t^-$ to sharpen/flatten the conditional distributions. Based on our observations, when $t^+ = 3$ and $t^- = 3$, the conditional distributions are dominated by 1-2 samples, where CACR can be regarded as the above-mentioned triplet loss, and this triplet loss with hard mining slightly underperforms CACR. From these views, CACR provides a more general form to connect the triplet loss. Meanwhile, it is interesting to notice CACR explains how triplet loss is deployed in the self-supervised learning scenario.

**Relation to CT**. The CT framework of Zheng & Zhou (2021) is primarily focused on measuring the difference between two different distributions, which are referred to as the source and target distributions, respectively.

It defines the expected CT cost from the source to target distributions as the forward CT, and that from the target to source as the backward CT. Minimizing the combined backward and forward CT cost, the primary goal is to optimize the target distribution to approximate the source distribution with both mode-covering and mode-seeking properties. In CACR, we did not find any performance boost by modeling the reverse conditional transport, since the marginal distributions of $\boldsymbol{x}$ and $\boldsymbol{x}^+$ are the same and these of $\boldsymbol{x}$ and $\boldsymbol{x}^-$ are also the same, there is no need to differentiate the transporting directions. In addition, the primary goal of CACR is not to regenerate the data but to learn $f_{\boldsymbol{\theta}}(\cdot)$ that can provide good latent representations for downstream tasks.

## C   Experiment details

On small-scale datasets, all experiments are conducted on a single GPU, including NVIDIA 1080 Ti and RTX 3090; on large-scale datasets, all experiments are done on 8 Tesla-V100-32G GPUs.

### C.1   Small-scale datasets: CIFAR-10, CIFAR-100, and STL-10

For experiments on CIFAR-10, CIFAR-100, and STL-10, we use the following configurations:

- **Data Augmentation**: We strictly follow the standard data augmentations to construct positive and negative samples introduced in prior works in contrastive learning (Wu et al., 2018; Tian et al., 2019; Hjelm et al., 2018; Bachman et al., 2019; Chuang et al., 2020; He et al., 2020; Wang & Isola, 2020). The augmentations include image resizing, random cropping, flipping, color jittering, and gray-scale conversion. We provide a Pytorch-style augmentation code in Algorithm 1, which is exactly the same as the one used in Wang & Isola (2020).

---

**Algorithm 1** PyTorch-like Augmentation Code on CIFAR-10, CIFAR-100 and STL-10

```python
import torchvision.transforms as transforms

# CIFAR-10 Transformation
def transform_cifar10():
    return transforms.Compose([
        transforms.RandomResizedCrop(32, scale=(0.2, 1)),
        transforms.RandomHorizontalFlip(),# by default p=0.5
        transforms.ColorJitter(0.4, 0.4, 0.4, 0.4),
        transforms.RandomGrayscale(p=0.2),
        transforms.ToTensor(), # normalize to value in [0,1]
        transforms.Normalize(
            (0.4914, 0.4822, 0.4465),
            (0.2023, 0.1994, 0.2010),
        )
    ])

# CIFAR-100 Transformation
def transform_cifar100():
    return transforms.Compose([
        transforms.RandomResizedCrop(32, scale=(0.2, 1)),
        transforms.RandomHorizontalFlip(),# by default p=0.5
        transforms.ColorJitter(0.4, 0.4, 0.4, 0.4),
        transforms.RandomGrayscale(p=0.2),
        transforms.ToTensor(), # normalize to value in [0,1]
        transforms.Normalize(
            (0.5071, 0.4867, 0.4408),
            (0.2675, 0.2565, 0.2761),
        )
    ])

# STL-10 Transformation
def transform_stl10():
    return transforms.Compose([
        transforms.RandomResizedCrop(64, scale=(0.08, 1)),
        transforms.RandomHorizontalFlip(),# by default p=0.5
        transforms.ColorJitter(0.4, 0.4, 0.4, 0.4),
        transforms.RandomGrayscale(p=0.2),
        transforms.ToTensor(), # normalize to value in [0,1]
        transforms.Normalize(
            (0.4409, 0.4279, 0.3868),
            (0.2683, 0.2610, 0.2687),
        )
    ])
```

---

- **Feature Encoder**: Following the experiments in Wang & Isola (2020), we use an AlexNet-based encoder as the feature encoder for these three datasets, where encoder architectures are the same

Table 21: The 100 randomly selected classes from ImageNet forms the ImageNet-100 dataset. These classes are the same as (Wang & Isola, 2020; Tian et al., 2019).

| ImageNet-100 Classes | | | | | | | | | |
|---|---|---|---|---|---|---|---|---|---|
| n02869837 | n01749939 | n02488291 | n02107142 | n13037406 | n02091831 | n04517823 | n04589890 | n03062245 | n01773797 |
| n01735189 | n07831146 | n07753275 | n03085013 | n04485082 | n02105505 | n01983481 | n02788148 | n03530642 | n04435653 |
| n02086910 | n02859443 | n13040303 | n03594734 | n02085620 | n02099849 | n01558993 | n04493381 | n02109047 | n04111531 |
| n02877765 | n04429376 | n02009229 | n01978455 | n02106550 | n01820546 | n01692333 | n07714571 | n02974003 | n02114855 |
| n03785016 | n03764736 | n03775546 | n02087046 | n07836838 | n04099969 | n04592741 | n03891251 | n02701002 | n03379051 |
| n02259212 | n07715103 | n03947888 | n04026417 | n02326432 | n03637318 | n01980166 | n02113799 | n02086240 | n03903868 |
| n02483362 | n04127249 | n02089973 | n03017168 | n02093428 | n02804414 | n02396427 | n04418357 | n02172182 | n01729322 |
| n02113978 | n03787032 | n02089867 | n02119022 | n03777754 | n04238763 | n02231487 | n03032252 | n02138441 | n02104029 |
| n03837869 | n03494278 | n04136333 | n03794056 | n03492542 | n02018207 | n04067472 | n03930630 | n03584829 | n02123045 |
| n04229816 | n02100583 | n03642806 | n04336792 | n03259280 | n02116738 | n02108089 | n03424325 | n01855672 | n02090622 |

as those used in the corresponding experiments in Tian et al. (2019) and Wang & Isola (2020). Moreover, we also follow the setups in Robinson et al. (2021) and test the performance of CACR with a ResNet50 encoder (results are shown in Table 9).

- **Model Optimization**: We apply the mini-batch SGD with 0.9 momentum and 1e-4 weight decay. The learning rate is linearly scaled as 0.12 per 256 batch size (Goyal et al., 2017). The optimization is done over 200 epochs, and the learning rate is decayed by a factor of 0.1 at epoch 155, 170, and 185.

- **Parameter Setup**: On CIFAR-10, CIFAR-100, and STL-10, we follow Wang & Isola (2020) to set the training batch size as $M = 768$ for baselines. The hyper-parameters of CL, AU-CL⋆, and HN-CL⋆ are set according to the original paper or online codes. Specifically, the temperature parameter of CL is $\tau = 0.19$, the hyper-parameters of AU-CL are $t = 2.0, \tau = 0.19$, and the hyper-parameter of HN-CL are $\tau = 0.5, \beta = 1.0$⋆, which shows the best performance according to our tuning. For CMC and CACR with multiple positives, the positive sampling size is $K = 4$. To make sure the performance is not improved by using more samples, the training batch size is set as $M = 128$. For CACR, in both single and multi-positive sample settings, we set $t^+ = 1.0$ for all small-scale datasets. As for $t^-$, for CACR ($K = 1$), $t^-$ is 2.0, 3.0, and 3.0 on CIFAR-10,CIFAR100, and STL-10, respectively. For CACR ($K = 4$), $t^-$ is 0.9, 2.0, and 2.0 on CIFAR-10, CIFAR100, and STL-10, respectively. For further ablation studies, we test $t^+$ and $t^-$ with the search in the range of $[0.5, 0.7, 0.9, 1.0, 2.0, 3.0]$, and we test all the methods with several mini-batch sizes $M \in \{64, 128, 256, 512, 768\}$.

- **Evaluation**: The feature encoder is trained with the default built-in training set of the datasets. In the evaluation, the feature encoder is frozen, and a linear classifier is trained and tested on the default training set and validation set of each dataset, respectively. Following Wang & Isola (2020), we train the linear classifier with Adam optimizer over 100 epochs, with $\beta_1 = 0.5$, $\beta_2 = 0.999$, $\epsilon = 10^{-8}$, and 128 as the batch size. The initial learning rate is 0.001 and decayed by a factor of 0.2 at epoch 60 and epoch 80. Extracted features from "fc7" are employed for the evaluation. For the ResNet50 setting in Robinson et al. (2021), the extracted features are from the encoder backbone with dimension 2048.

## C.2 Large-scale datasets

For large-scale datasets, the Imagenet-1K is the standard ImageNet dataset that has about 1.28 million images of 1000 classes. The ImageNet-100 contains randomly selected 100 classes from the standard ImageNet-1K dataset, and the classes used here are the same with Tian et al. (2019) and Wang & Isola (2020), listed in Table 21. For pretraining with less curated data⋆, considering Webvision v1 and ImageNet-22k respectively have 2.4 and 14.2 million images, we decrease the training epoch to 1/2 and 1/10 when pretraining on these two datasets as done in Li et al. (2021). We follow the standard settings in these works and describe the experiment configurations as follows:

---

⋆https://github.com/SsnL/align_uniform

⋆https://github.com/joshr17/HCL

⋆Please refer to the original paper for the specific meanings of the hyper-parameter in baselines.

⋆https://data.vision.ee.ethz.ch/cvl/webvision/dataset2017.html

---

**Algorithm 2** PyTorch-like Augmentation Code on ImageNet-100 and ImageNet-1K

---

```python
import torchvision.transforms as transforms
# ImageNet-100 and ImageNet-1K Transformation
# MoCo v2's aug: similar to SimCLR https://arxiv.org/abs/2002.05709
def transform_imagenet():
    return transforms.Compose([
        transforms.RandomResizedCrop(224, scale=(0.2, 1.)),
        transforms.RandomApply([transforms.ColorJitter(0.4, 0.4, 0.4, 0.1)
                    ], p=0.8),
        transforms.RandomGrayscale(p=0.2),
        transforms.RandomApply([moco.loader.GaussianBlur([.1, 2.])], p=0.5),
        transforms.RandomHorizontalFlip(),
        transforms.ToTensor(),
        transforms.Normalize(mean=[0.485, 0.456, 0.406],
                std=[0.229, 0.224, 0.225])
    ])
```

---

- **Data Augmentation**: Following Tian et al. (2019; 2020b); Wang & Isola (2020); Chuang et al. (2020); He et al. (2020), and Chen et al. (2020c), the data augmentations are the same as the standard protocol, including resizing, 1x image cropping, horizontal flipping, color jittering, and gray-scale conversion with specific probability. The full augmentation combination is shown in Algorithm 2.

- **Feature Encoder**: For the main experiments, we apply MoCo-v3 setting, where the framework consists of a teacher and a student encoder. The teacher encoder follows a EMA updating strategy and all experiment settings follows the MoCo-v3 paper (Chen et al., 2021). We apply the MoCo-v2 framework (Chen et al., 2020c) for further justification that CACR is also applicable with framework including a large queue, where the ResNet50 (He et al., 2016) is a commonly chosen feature encoder architecture. The output dimension of the encoder is set as 128.

- **Model Optimization**: Following the standard setting in He et al. (2020); Chen et al. (2020c); Wang & Isola (2020), the training mini-batch size is set as 128 on ImageNet-100 and 256 on ImageNet-1K. We use a mini-batch stochastic gradient descent (SGD) optimizer with 0.9 momentum and 1e-4 weight decay. The total number of training epochs is set as 200. The learning rate is initialized as 0.03, decayed by a cosine scheduler for MoCo-V2 at epoch 120 and epoch 160. In all experiments, the momentum of updating the offline encoder is 0.999.

- **Parameter Setup**: On ImageNet-100 and ImageNet-1K, for all methods, the queue size for negative sampling is 65,536. The training batch size is 128 on ImageNet-100 and 256 on ImageNet. For CACR, we train with two positive sampling sizes $K = 1$ and $K = 4$ and the parameters in the conditional weight metric are respectively set as $t^+ = 1.0$, $t^- = 2.0$. For baselines, according to their papers and Github pages (Tian et al., 2019; Wang & Isola, 2020; Robinson et al., 2021), the temperature parameter of CL is $\tau = 0.2$, the hyper-parameters of AU-CL are $t = 3.0$, $\tau = 0.2$, and the hyper-parameters of HN-CL are $\tau = 0.5$, $\beta = 1.0$. Note that CMC ($K = 1$) reported in the main paper is trained with 240 epochs and with its own augmentation methods (Tian et al., 2019). For CMC ($K = 4$), the temperature is set $\tau = 0.07$ according to the setting in Tian et al. (2019) and the loss is calculated with Equation (8) in the paper, which requires more GPU resources than 8 Tesla-V100-32G GPUs with the setting on ImageNet-1K.

- **Linear Classification Evaluation**: Following the standard linear classification evaluation (He et al., 2020; Chen et al., 2020c; Wang & Isola, 2020), the pre-trained feature encoders are fixed, and a linear classifier added on top is trained on the train split and test on the validation split. The linear classifier is trained with SGD over 100 epochs, with the momentum as 0.9, the mini-batch size as 256, and the learning rate as 30.0, decayed by a factor of 0.1 at epoch 60 and epoch 80.

- **Feature Transferring Evaluation (Detection and Segmentation)**: The pre-trained models are transferred to various tasks including PASCAL VOC* and COCO* datsets. Strictly following the

---

*http://host.robots.ox.ac.uk/pascal/VOC/index.html
*https://cocodataset.org/#download

same setting in He et al. (2020), for the detection on Pascal VOC, a Faster R-CNN (Ren et al., 2016) with an R50-C4 backbone is first fine-tuned end-to-end on the VOC 07+12 trainval set and then evaluated on the VOC 07 test set with the COCO suite of metrics (Lin et al., 2014). The image scale is [480, 800] pixels during training and 800 at inference. For the detection and segmentation on COCO dataset, a Mask R-CNN (He et al., 2017) with C4 backbone (1x schedule) is applied for the end-to-end fine-tuning. The model is tuned on the train2017 set and evaluate on val2017 set, where the image scale is in [640, 800] pixels during training and is 800 in the inference.

- **Image-Text representation learning**: Following the CLIP setting (Radford et al., 2021), we adopt ResNet50 as the vision encoder backbone. All methods are pre-trained for 32 epochs, with the mini-batch size as 2048, and the learning rate as 5e-4, with cosine annealing scheduler.

- **Estimation of $\hat{\pi}_{\boldsymbol{\theta}}^-$ with MoCo-v2:** Following the strategy in Wang & Isola (2020), we estimate $\hat{\pi}_{\boldsymbol{\theta}}^-$ with not only the cost between queries and keys, but also with the cost between queries. Specifically, at each iteration, let $M$ be the mini-batch size, $N$ be the queue size, $\{f_{q_i}\}_{i=1}^M$ be the query features, and $\{f_{k_j}\}_{j=1}^N$ be the key features. The conditional distribution is calculated as:

$$\hat{\pi}_{\boldsymbol{\theta}}^-(\cdot|f_{q_i}) = \frac{e^{-d_{t^-}(\cdot, f_{q_i})}}{\sum_{j=1}^N e^{-d_{t^-}(f_{k_j}, f_{q_i})} + \sum_{j \neq i} e^{-d_{t^-}(f_{q_j}, f_{q_i})}}$$

To be clear, the Pytorch-like pseudo-code is provided in Algorithm 3. In MoCo-v2 framework, as the keys are produced from the momentum encoder, this estimation could help the main encoder get involved with the gradient from the conditional distribution, which is consistent with the formulation in Section 3.2.

---

**Algorithm 3** PyTorch-like style pseudo-code of CACR with MoCo-v2 at each iteration.

---

```
####################### Inputs #######################
# t_pos, t_neg: hyper-parameters in CACR
# m: momentum
# im_list=[B0, B2, ..., BK] list of mini-batches of length (K+1)
# B: mini-batches (Mx3x224x224), M denotes batch_size
# encoder_q: main encoder; encoder_k: momentum encoder
# queue: dictionary as a queue of N features of keys (dxN); d denotes the feature dimension

######## compute the embeddings of all samples ################
q_list = [encoder_q(im) for im in im_list] # a list of (K+1) queries q: M x d
k_list = [encoder_k(im) for im in im_list]
stacked_k = torch.stack(k_list, dim=0) # keys k: (K+1) x M x d

################## compute the loss ###################
CACR_loss_pos, CACR_loss_neg = 0.0, 0.0
for k in range(len(im_list)): #load a mini-batch with M samples as queries
    q = q_list[k]
    mask = list(range(len(im_list)))
    mask.pop(k) # the rest mini-batches are used as positive and negative samples

    #################### compute the positive cost ####################
    # calculate the cost of moving positive samples: M x K
    cost_for_pos = (q - stacked_k[mask]).norm(p=2, dim=-1).pow(2).transpose(1, 0) # point-to-point cost
    with torch.no_grad(): # the calculation involves momentum encoder, so with no grad here.
        # calculate the conditional distribution: M x K
        weights_for_pos = torch.softmax(cost_for_pos.mul(t_pos), dim=1) # calculate the positive conditional distribution
    # calculate the positive cost with the empirical mean
    CACR_loss_pos += (cost_for_pos*weights_for_pos).sum(1).mean()

    #################### compute the loss of negative samples #####################
    # calculate the cost and weights of negative samples from the queue: M x K
    sq_dists_for_cost = (2 - 2 * mm(q, queue))
    sq_dists_for_weights = sq_dists_for_cost

    if with_intra_batch: # compute the distance of negative samples in the mini-batch: Mx(M-1)
        intra_batch_sq_dists = torch.norm(q[:,None] - q, dim=-1).pow(2).masked_select(~torch.eye(q.shape[0], dtype=bool).
            cuda()).view(q.shape[0], q.shape[0] - 1)
        # combine the distance of negative samples from the queue and intra-batch: Mx(K+M-1)
        sq_dists_for_cost = torch.cat([sq_dists_for_cost, intra_batch_sq_dists], dim=1)
        sq_dists_for_weights = torch.cat([sq_dists_for_weights, intra_batch_sq_dists], dim=1)

    # calculate the negative conditional distribution: if with_intra_batch==True Mx(K+M-1), else MxK
    weights_for_neg = torch.softmax(sq_dists_for_weights.mul(-t_neg), dim=1)
    # calculate the negative cost with the empirical mean
    CACR_loss_neg += (sq_dists_for_cost.mul(-1.0)*weights_for_neg).sum(1).mean()

# combine the loss of positive cost and negative cost and update main encoder
CACR_loss = CACR_loss_pos/len(im_list)+CACR_loss_neg/len(im_list)
CACR_loss.backward()
update(encoder_q.params) # SGD update: main encoder
encoder_k.params = m*encoder_k.params+(1-m)*encoder_q.params #momentum update: key encoder

# update the dictionary, dequeue and enqueue
enqueue(queue, k_list[-1]) # enqueue the current minibatch
dequeue(queue) # dequeue the earlist minibatch
```

---

pow: power function; `mm`: matrix multiplication; `cat`: concatenation.

