# OpenReview forum: "Contrastive Attraction and Contrastive Repulsion for Representation Learning"
_TMLR — Accepted by TMLR_

### Review · Reviewer_SXhF · 2023-05-14

**Summary Of Contributions:**

This work extends the general contrastive learning (CL) algorithms by sampling more positive examples during training. The authors first use theoretic analysis to motivate the idea of separating repulsion and aggregation in the loss formulation. They then use empirical studies showing that the proposed CACR algorithm consistently outperforms current SOTA algorithms and yield significant improvement on few-shot image classification task on dataset ELEVATER.

**Audience:**

Yes

**Broader Impact Concerns:**

No clear negative society impacts from this. The authors have addressed this in their paper.

**Claims And Evidence:**

No

**Requested Changes:**

See the weakness.

**Strengths And Weaknesses:**

Strengths:
1.	The algorithm is well motivated, as it is intuitive to think that the single positive example typically used by CL algorithms should not be optimal. And indeed, slightly earlier proposed algorithm SwAV already showed that adding multiple small-crops as positive examples during training helps boost the performance. So this work is another positive support to this idea, with some minor difference in how the positive examples are selected and used.
2.	The experiment results are good. CACR outperforms most of the existing algorithms and surpasses the MoCo-v3 by large margins on the ELEVATER benchmark.
3.	The paper is in general clearly written. It also provides a lot of details in the appendix.

Weaknesses:
1.	Although the paper is mostly written clearly, the algorithm itself is not clearly explained. Especially, some key details are not consistent between the formulas and (some of) the code. For example, the equation for loss of CA at the bottom of page 5 seems to indicate that there is a “central” representation for “x_i”, which I don’t think is actually happening as most recent contrastive learning algorithms have used the augmentation examples as positive examples without a central representation (like what was stored in memory bank in Wu et al. 2018). It is also unclear to me how the augmentations of the other examples within the same batch are used to compute the loss of CR, if the algorithm uses SimCLR style loss, they would also be treated as the negative examples, but that’s inconsistent to what was described in the equation for CR. This becomes especially confusing when the section of the large-scale datasets starts. The authors say that “following the convention, we adapt CACR loss to the MoCo-v2 framework”. What convention is followed? What does it mean by “adapting CACR loss to the MoCo-v2 framework”?
2.	Although the performance gain on the ELEVATER dataset is very significant, it feels a little bit too good to make people feel that the MoCo-v3 baseline may not be carefully done. This worry is really amplified by the relatively small performance gains on the other datasets. I feel the authors need to justify more about whether the MoCo-v3 baseline is carefully done, or to provide more model results, or to explain why these gains are so large in more convincing way (like how the gains will be built up by estabilishing the special mechanisms proposed by the authors).
3.	The citation for the convention CL algorithm (Oord et al. 2018) seems to be inaccurate. The Oord et al. 2018 is a contrastive predictive coding method, which is not a contrastive learning method. I would not think this method can achieve so good performance on the reported datasets. Did the author mean the SimCLR paper?

---

> ### Author Response · Authors · 2023-06-02
> **Response to Reviewer SXhF**
>
> We appreciate your review and comments. Please find our responses below:
>
> > The algorithm itself is not clearly explained. Does CACR use SimCLR style loss and how to handle “central” representations? How to adapt CACR loss into MoCo-v2 framework?
>
> We understand that a more detailed explanation of the CACR method would be beneficial and have provided this in Appendix C.2, which includes Pytorch-like pseudo-code for ease of understanding.
>
> The framework of SimCLR style methods (as well as others like MoCo and SimSiam) typically uses an encoder to handle two augmentations, each serving as the central representation for the other. Mathematically, if we consider a pair of positive augmentations, $x_1$ and $x_2$, when $x_1$ acts as the central representation, $x_2$ is treated as its positive sample, $x^+$, and vice versa. The loss is then computed as an average across these two augmentations.
>
> In the context of CACR, we extend this idea to include K augmentations, $x_1, x_2, …, x_K$. Each of these can serve as the central representation, while the remaining representations are used as positive samples. The loss is then calculated as an average over each augmentation.
>
> For adapting the CACR loss into the MoCo-v2 framework, we follow the approach outlined in Wang and Isola (2020), which adapts their Align and Uniform loss to the MoCo-v2 framework. In this setup, the CR loss changes slightly as we need to incorporate the negative samples stored in the momentum queue to compute the conditional distribution. Other than this modification, the rest of the setup remains consistent with our normal settings.
>
> We have revised our paper to incorporate these explanations for better understanding of the CACR algorithm.
>
> >  Although the performance gain on the ELEVATER dataset is very significant, it feels a little bit too good to make people feel that the MoCo-v3 baseline may not be carefully done. This worry is really amplified by the relatively small performance gains on the other datasets.
>
> In the ELEVATER experiments, it's important to note that we followed the benchmark configurations provided by the ELEVATER study precisely. The numbers reported in our paper are taken directly from Table 12 in the ELEVATER paper (Li et al., 2023), where a grid search was applied to identify the best hyper-parameter settings. We also examined the results and found that the reported numbers represent the best performance, suggesting that the comparison is indeed fair.
>
> The goal of this study was not just to achieve high performance on the datasets used during pre-training, such as ImageNet1K, but also to demonstrate the generalization ability of our approach to other datasets. We believe that this is a critical aspect of developing robust and reliable models, and the ELEVATER benchmark provides a nice opportunity to evaluate this aspect. Moreover, on other datasets like CIFAR, STL, the performance gains are also significant.
>
> > The citation for the convention CL algorithm (Oord et al. 2018) seems to be inaccurate in Table 1.
>
> Thanks for pointing this out. We have fixed it in the revision.
>
>
> ---------------------------------
>
> We hope our responses address your concerns and are open to further questions or comments.

---

### Review · Reviewer_3T1n · 2023-05-19

**Summary Of Contributions:**

The authors propose a contrastive learning approach that decomposes the loss into an attraction and repulsion term and re-weight hard positives and hard negatives. An advantage of such procedure is that it puts more weight on hard negatives, which makes it more robust on imbalanced distributions. The authors provide a theoretical analysis based on Wang and Isola's work on uniformity and alignment and empirical results comparing their method to other robust contrastive learning approaches such as Huynh et al. 2020 and AU-CL. The authors provide an extensive appendix with proofs, additional results and analysis, an ablation, GPU times, experiment details, and algorithms.

**Audience:**

Yes

**Broader Impact Concerns:**

The current broader impact is enough.

**Claims And Evidence:**

Yes

**Requested Changes:**

* Fix minor issues


**Strengths And Weaknesses:**

Strengths
=======
* The proposed method is sound
* Experiments show that the proposed method is more robust to data imbalances and noise than MocoV2.
* The authors provide ablations, details to reproduce their results, and a theoretical analysis.
* The authors provide experiments on a large number of benchmarks.
* In general the paper is well-written
* The authors acknowledge that their method could fail in the presence of false negatives.

Weaknesses
=========
* Although CACR improves robustness, results still show a big drop in performance on imbalanced settings, close to methods not using CACR.
* Hard-negative reweighing is a known technique in the metric learning literature [A], and, as the authors acknowledge in the related work, hard negative mining has been explored in SSL [B, C].

[A] Wang, Xun, et al. "Multi-similarity loss with general pair weighting for deep metric learning." Proceedings of the IEEE/CVF conference on computer vision and pattern recognition. 2019.
[B] Tabassum, Afrina, et al. "Hard negative sampling strategies for contrastive representation learning." arXiv preprint arXiv:2206.01197 (2022).
[C] Xu, Lanling, et al. "Negative Sampling for Contrastive Representation Learning: A Review." arXiv preprint arXiv:2206.00212 (2022).

Minor issues
==========
* I find this sentence a bit complicated to read: "we propose a doubly CL strategy that contrasts positive samples and negative ones within themselves separately."
* "Eqn. equation 4 and p(x+ | xdata ) for Eqn. equation 3 with x+ = T (xdata , ε )" -> Repeated Eqn. equation
* Lemma 4.3: "become the Uniformity" -> becomes
* "the lack of intra-positive contrast shows the gap of repulsion efficiency" -> attraction efficiency?

---

> ### Author Response · Authors · 2023-06-02
> **Response to Reviewer 3T1n**
>
> Thank you for your feedbacks and suggestions, please find our responses to your concerns below:
>
> > Although CACR improves robustness, results still show a big drop in performance on imbalanced settings, close to methods not using CACR.
>
> While it is true that there is a performance drop in imbalanced settings, it's important to contextualize this drop. This decrease in performance is largely due to the inherent difficulty of learning from imbalanced class distributions present in the pre-training dataset, a challenge faced by any model in this situation.
>
> Although the performance drop in CACR is similar to that of the baselines, the absolute performance of CACR remains consistently higher than the baselines. This is because CACR demonstrates superior performance on balanced datasets, and this advantage is preserved even in the face of imbalance.
>
> Furthermore, it's worth noting that CACR tends to exhibit a less dramatic performance drop than the baselines in many imbalanced scenarios. This is particularly true for both small-scale data like CIFAR-100 and large-scale data like ImageNet-22K. Thus we believe these points illustrate the robustness of CACR in different learning conditions, despite the universally challenging nature of imbalanced settings.
>
> > Hard-negative reweighing is a known technique in the metric learning literature [A], and, as the authors acknowledge in the related work, hard negative mining has been explored in SSL [B, C].
>
> We appreciate your comments and have added the missed reference in our revision. It's true that the principle of hard-negative reweighing is a known technique in these fields. We acknowledge these influences. The key novelty of CACR lies in its doubly-contrastive approach to representation learning, which is designed to capture distinct information from both positive and negative samples with respect to the query sample. This is achieved through the formulation of intra-sample contrasts using conditional probability. In this context, the conditional probability in the Contrastive Repulsion (CR) component of our method does exhibit hard negative mining properties, and in that sense, there is a connection to the works you cited [A-C]. However, the key distinction lies in the fact that CACR is not purely driven by hard-negative mining like the techniques in these previous works. Beyond this point, it also leverages a combination of both hard-positive and hard-negative mining principles, with the ultimate goal of enhancing contrastive learning. Please also refer our response to the first question of Review co9r.
>
> > Minor issues
>
> We appreciate you pointing them out. They are fixed in our revision.
>
> --------------------------
> Thanks for your comments. Your feedback is valuable, and we welcome any further comments or queries.

---

### Review · Reviewer_p4Wu · 2023-05-23

**Summary Of Contributions:**

This paper studies contrastive representation learning and points out that the current literature merely considered the intra-relation between samples, leading to unsatisfactory performance. To address this, this paper introduces a doubly contrastive strategy called CACR that can attract positive samples and repel negative samples separately. Finally, experiments are conducted to validate the effectiveness of the proposed method.

**Audience:**

Yes

**Claims And Evidence:**

Yes

**Requested Changes:**

Please consider my concerns first in Strengths & Weaknesses.

1. According to Wang&Isola (2020), it is better to conduct the Alignment and Uniformity analysis to show the effectiveness of the proposed approach.
2. In Sec.3.2, the number of M seems essential to the approximate distribution performance. Therefore, it is necessary to make an ablation study about this.

**Strengths And Weaknesses:**

The targeting problem "contrastive representation learning" is an essential topic in the machine learning community, and the proposed algorithm seems reasonable. Moreover, the writing is easy to follow.

However, there are also some critical defects in this work.

1. The survey along the contrastive representation learning is insufficient, and some advanced/sota methods are not considered/discussed in both related work and experiments, such as [1,2,3,4].
2. The theoretical analysis in Sec.4 is limited and needs further discussion. Specifically, a) In Lem.4.1, it is non-trivial to make sure $\pi_{\theta}^{+}=[(x^+|x_0)$. b) It seems that we have to let $c(z_1,z_2)=d(z_1, z_2)$ to derive Lem.4.2 and Lem.4.3, which means that both $t^+$ and $t^-$ should be $1.0$ in Eqn.(3) and Eqn.(4). Yet CACR cannot achieve promising performance in this case as shown in Tab.12 and Tab.13. c) In Sec.4.1, $x$ and $x^+$ are not strictly independent because they are obtained by transforming from the same sample $x_0$ as described in Sec.3.
3. The experiments are unconvincing because some of the sota methods are not included, such as [1,2,3,4].
4. Some of the notions are not introduced/explained. For example, in Eqn.(5), $\delta_{x_j}$ and $\delta_{x_{ik}^+}$. In Lem.4.3, is the M same as that one in Eqn. (5)?


Ref:
1. ADAPTIVE CONTRASTIVE LEARNING OF REPRESENTATION BY NEAREST POSITIVE EXPANSION.
2. Adaptive Soft Contrastive Learning.
3. Intra- and Inter-Contrastive Learning for Micro-expression Action Unit Detection.
4. Parametric Contrastive Learning.

---

> ### Author Response · Authors · 2023-06-02
> **Response to Reviewer p4Wu**
>
> We appreciate your valuable comments. Please find our point-to-point responses regarding your concerns:
>
> > The survey along the contrastive representation learning is insufficient, and some advanced/sota methods are not considered/discussed in both related work and experiments, such as [1,2,3,4]
>
> We agree that our paper can be strengthened by discussing these advancements in contrastive representation learning. In our revision, we have included these references and established their connection to CACR. To briefly summarize, both [1] and [2] approach the task by treating samples unequally, similar to our approach. [1] selects the top-k important samples based on feature similarity, while [2] defines pseudo-labels using a conditional distribution similar to ours. [3] employs a blend of intra- and inter-contrastive loss, which aligns with our discussion in Appendix B.5. Finally, [4] applies self-supervised contrastive learning loss in supervised settings.
>
> We've also added the results from [1,2] in Table 4 of the revised paper, providing a direct comparison with CACR. As for [3] and [4], we noted that they don't provide results in the standard self-supervised learning evaluation setting, or they evaluate performance using end-to-end training instead of linear probing. Therefore, a direct comparison is not applicable.
>
> In terms of linear probing performance, the table below provides a comparison between CACR, ADACLR [1], and ASCL [2]. As can be seen, CACR outperforms both ADACLR and ASCL:
>
> | Model  | Top1 Accuracy |
> | ------ | ------------- |
> | ADACLR | 72.3%         |
> | ASCL   | 71.5%         |
> | CACR   | 74.4%         |
>
> > It is non-trivial to make sure $\pi_\theta^+ = p(x^+| x_0)$
>
> Thank you for your observation. The way of treating $\pi_\theta^+ $ is a key distinction between our approach, CACR, and traditional contrastive learning (CL) methods. In CACR, we move away from the typical approach of treating positive samples as independent with respect to the query sample.
>
> In Section 4.1 of our paper, we discuss how $\pi_\theta^+(x^+ | x, x_0) = p(x^+| x_0)$ is only true when the model has been optimized. Empirical evidence supporting this point can be found in Table 11 of our paper.
>
> When comparing the performance of CACR and CACR w/o $\pi_\theta^+$ (where w/o $\pi_\theta^+$ means that we set $\pi_\theta^+ = p(x^+| x_0)$ by default), it's clear that the overall performance of CACR consistently surpasses that of CACR w/o $\pi_\theta^+$. This validates our approach of dynamically determining $\pi_\theta^+$ during optimization, rather than setting it equal to a uniform $p(x^+| x_0)$ by default.
>
> > It seems that we have to let $c(z_1,z_2)= d(z_1,z_2)$  to derive Lem.4.2 and Lem.4.3, which means that both t+ and t− should be 1.0 in Eqn.(3) and Eqn.(4)
>
> Thank you for your observation. However, our derivations for Lemma 4.2 and Lemma 4.3 do not necessitate the assumption that $c(z_1,z_2)= d(z_1,z_2)$. As detailed in Appendix A, the proofs for these lemmas are constructed without the need for this specific condition. Instead, we introduce the condition $c(z_1, z_2)= - z_1^\top z_2 = - f_\theta(x_1)^\top f_\theta(x_2)$ specifically when drawing parallels to the loss function used in traditional contrastive learning (CL) approaches. This distinction allows us to maintain a broader scope in our theoretical framework, while still enabling us to make important connections to conventional CL methods when appropriate.
>
> > In Sec.4.1, x and x_+ are not strictly independent because they are obtained by transforming from the same sample x_0 as described in Sec.3.
>
> You're correct, and we appreciate you pointing out this important detail. In our setting, $x$ and $x_+$ are indeed not strictly independent because they both originate from transformations of the same root sample $x_0$, as described in Section 3. This underlying dependency is one of the key motivations behind our approach in CACR. Unlike traditional contrastive learning (CL) methods, we argue against setting $\pi_\theta^+ = p(x^+| x_0)$ precisely because of this dependency.
>
> Instead, we propose a framework that acknowledges and leverages this relationship. We believe this approach more accurately reflects the data generation process and can lead to more effective learning.
>
> > The experiments are unconvincing because some of the sota methods are not included, such as [1,2,3,4].
>
> As per your first question, we have added the comparison with the state-of-the-art methods you mentioned [1,2,3,4] in our revision. We have not only discussed their connection to our method in the revised manuscript but also included their results in our experimental section for a direct comparison. This comparison demonstrates how our method, CACR, performs in relation to these recent advancements in the field. We believe these additions make our experimental evaluation more comprehensive and convincing. Please refer to our response to your first question for more details.

---

> > ### Author Response · Authors · 2023-06-02
> > **Response to Reviewer p4Wu (part 2)**
> >
> > > Some of the notions are not introduced/explained. For example, in Eqn.(5), $\delta_{x_j}$ and $\delta_{x_{ik^+}}$. In Lem.4.3, is the M same as that one in Eqn. (5)?
> >
> > We apologize for any lack of clarity in the paper. In Equation (5), $\delta_{x_j}$ and $\delta_{x_{ik^+}}$ are Dirac delta functions centered at $x_j$ and $x_{ik^+}$, respectively. These functions are used to represent the empirical distribution of the data.
> >
> > As for the notation “M”, you're correct. The "M" in Lemma 4.3 is the same as the "M" in Equation (5). In both cases, "M" represents the batch size.
> >
> > >  It is better to conduct the Alignment and Uniformity analysis to show the effectiveness of the proposed approach.
> >
> > We agree that the Alignment and Uniformity analysis can provide valuable insights into the effectiveness of the proposed method. As a matter of fact, our analysis in Lemma 4.1 and 4.2 is conceptually aligned with the Alignment and Uniformity analysis described in Wang and Isola (2020).
> >
> > More specifically, our approach mirrors the alignment and uniformity objectives by ensuring that positive pairs align closely in the embedding space (Alignment) while the embeddings of negative pairs are uniformly distributed (Uniformity). For a more detailed discussion on the connection of our work to the Alignment and Uniformity analysis in Wang and Isola (2020), please refer to section B.5 in the appendix of our paper. Here we demonstrate that CACR with RBF cost and K=1 reduces to the align and uniform loss used in Wang and Isola (2020).
> >
> > > In Sec.3.2, the number of M seems essential to the approximate distribution performance. Therefore, it is necessary to make an ablation study about this.
> >
> > We do provide ablation studies regarding the size of M. Please see the results shown in Figure 8 and corresponding text in B.3 - “on the effects of negative sampling size”.
> >
> > ------------------------
> >
> > We appreciate your feedback and are open to further queries or comments.

---

### Review · Reviewer_co9r · 2023-05-28

**Summary Of Contributions:**

First, thank you to the authors. I enjoyed reading their work, and believe it has a number of merits.

Summary:

This paper presents a simple and effective contrastive leanring method for assigning increased importance weighting to positive samples that are far from the query and negative samples that are close to the query. In other words, the proposed method samples hard negative and hard positive sampling in contrastive learning.

Background:

The paper's contribution lies in proposing a unified approach to both hard positive and negative sampling. The method demonstrates promising results, and the message of the paper is simple and clear to understand.

However, I have concerns about the novelty of the paper. The negative sampling strategy (“contrastive repulsion”) is exactly the same probability distribution as [1] and should be cited accordingly in Sec 3.1. The main methodological contribution is the suggestion of simultaneously applying the same approach (with inverted weighting) to sample hard positives. Whilst this is reasonable, the conceptual contribution is marginal as other works also consider similar methods (see e.g., [2]).

The theory section is reasonable and correct, but not a significant contribution: the results are all standard.

References:

[1] Contrastive learning with hard negative samples, Robinson et al ICLR 2021.
[2] Unsupervised Representation Learning by Invariance Propagation, Wang et al. NeurIPS 2020.

**Audience:**

Yes

**Claims And Evidence:**

Yes

**Requested Changes:**

Add discussion of related hard negative and positive sampling approaches and cite in appropriate places (e.g., "contrastive repulsion" section 3.1).

**Strengths And Weaknesses:**

Strengths:
- The proposed method is easy to implement and sensible.
- Experimental evaluation is largely extensive enough (see caveat in weaknesses).
- It is encouraging to see that the method can effectively leverage multiple positive samples (K > 1).
- The paper is well-written, and the narrative is clear and easily understandable for readers.

Weaknesses:
- Concerns over novelty as discussed above.
- Experimental results are all of the flavor “our method beats the baselines on these benchmarks”. The paper would benefit from a closer study of the effect CACR has on the learned model (i..e, not benchmark evaluations, but experiments that probe the properties of the method).
- Why include the 5-shot fine-tune results in table 6? The networks have all overfitted and get worse performance than linear probe - why not just show linear probe

Conclusion:

I am loathe to reject papers due to lack of novelty, and shall not do so in this case. But I would like to register my dissatisfaction at the way the paper is written. It presents itself in a form in which it purports to present a “powerful new contrastive leanring method”, but the method is very similar to prior work, and the experimental results, whilst solid, are not enough to overcome this (e.g., 1.5% improvement when CACR is added to the MoCo-v2 method). My recommendation to the authors is to position it more as a study of the role of hard positive and negative samples, and position the paper more as an exploration of contrastive learning.

---

> ### Author Response · Authors · 2023-06-02
> **Response to Reviewer co9r**
>
> We appreciate you taking the time in reviewing the paper and enjoy the merits of our work. Please find our response regarding your concerns below:
>
> > Novelty of the paper and connection to HN-CL [1]  and to InvP-CL [2].
>
> Our work indeed builds upon previous research, including [1,2], and we acknowledge these influences. The key novelty of CACR lies in its doubly-contrastive approach to representation learning, which is designed to capture distinct information from both positive and negative samples with respect to the query sample. This is achieved through the formulation of intra-sample contrasts using conditional probability.
>
> Contrastive repulsion in our model employs a conditional probability that considers the relation between the query and negative samples. In contrast, the sampling strategy in [1] is influenced by both the $q_\beta^-$ (query-negative relation) and $q_\beta^+$ (query-positive relation). Therefore, while the two approaches may converge to a similar distribution when the number of negative samples approaches infinity (as we show in Lemma 4.3), they are not identical in finite-sample settings. In addition, [1] introduced two additional hyper-parameters $\tau^+$ and $\tau^-$ that need to tune on different datasets in order to define the sampling distribution ($q_\beta(x^-) = \tau^- q_\beta^-(x^-) + \tau^+ q_\beta^+(x^-)$), while the conditional distribution $\pi^-_\theta(x^-)$ in CR is more flexible to adapt different data and does not require further tuning.
>
> Furthermore, while our approach shares conceptual similarities with hard-sample mining strategies, we want to clarify that our primary motivation for CACR was not to mine hard samples. Rather, we sought to effectively incorporate diverse and informative sample contrasts in representation learning. We believe this nuanced approach is what sets our work apart from others.
>
> We appreciate your suggestion and will make sure to cite the relevant works in Section 3.1 to acknowledge their contribution to the field.
>
> > Experimental results are all of the flavor “our method beats the baselines on these benchmarks”. The paper would benefit from a closer study of the effect CACR has on the learned model (i..e, not benchmark evaluations, but experiments that probe the properties of the method)
>
> We appreciate your feedback and fully concur with your suggestion. Although our work has indeed demonstrated competitive performance on various benchmarks, we agree that the fundamental contribution of our study is the development and understanding of contrastive learning, not merely outperforming other methods.
>
> To this end, we have conducted an extensive analysis on the properties of CACR, particularly the impact of the conditional distribution in both the CA and CR components. We dive into the implications of these components and their effects on the representation learning in the appendix sections B.2 and B.3. These sections feature comprehensive discussions and experiments, including ablation studies on the design of $\pi^+$ and $\pi^-$ and analyses on the impact of sample size, among others.
>
> Our analysis supports the theoretical insights presented in Section 4, providing empirical evidence that complements our conceptual findings. We believe these discussions offer a thorough examination of the properties of our method, going beyond benchmark comparisons and shedding light on the inner workings of CACR.
>
> We hope this addresses your concern and we are open to further suggestions on how we can enhance our examination of CACR's properties.
>
> > Why include the 5-shot fine-tune results in table 6? The networks have all overfitted and get worse performance than linear probe - why not just show linear probe?
>
> The inclusion of 5-shot fine-tuning results in Table 6 is indeed to provide a complete view of our model's transferability, rather than focusing solely on superior performance. It's non-trivial to note that end-to-end fine-tuning and linear probing represent two prevalent strategies for adapting a pre-trained model to downstream tasks. We find that in 5-shot fine-tuning scenarios, linear probing tends to outperform fine-tuning, likely due to the model being heavily adapted to the pre-training data, thus making it less flexible for new tasks. However, it's noteworthy that CACR still exhibits relatively better transferability under these challenging conditions.
>
> As the amount of task-specific data increases, we observe that fine-tuning starts to outperform linear probing. To provide a more comprehensive picture of this trade-off, we have updated our paper to include full-shot results in Table 6. This addition provides a clearer understanding of how the performance of fine-tuning and linear probing evolves with the availability of more task-specific data.

---

> > ### Author Response · Authors · 2023-06-02
> > **Response to Reviewer co9r (part 2)**
> >
> >
> > > Add discussion of related hard negative and positive sampling approaches and cite in appropriate places (e.g., "contrastive repulsion" section 3.1).
> >
> > Thanks for your suggestions. We have added the citation accordingly and elaborated the relation discussion in the new section 3.3 (change the paragraph after section 3.2 “relation to typical CL loss” -> new section 3.3).
> >
> > ---------------
> >
> > We hope this addresses your concern and we are open to further suggestions on how we can enhance our examination of CACR's properties.

---

### Comment · Action_Editors · 2023-05-05
**Hi**

Dear all,

This paper is reassigned a new AE by the Chief Editor. I will do the following reviewing procedure. Thanks!

best,

AE

---

### Author Response · Authors · 2023-06-02
**Revision summary**

Dear AE and reviewers:

We greatly appreciate your insightful feedback. We've thoroughly revised our paper per your suggestions, marking all changes in blue for ease of reference. To further assist your review, we've summarized our revisions below:

**Abstract**

- We revise a hard-to-read sentence. (suggested by Reviewer 3T1n)

**Section2: Related work**

- We add missed references and corresponding discussions to these works. (suggested by Reviewer co9r, p4Wu, 3T1n)

**Section3: The proposed approach**

- We cite two suggested references in section 3.1 (suggested by Reviewer co9r)

- We clarify the explanation of the Dirac function $\delta$ in section 3.2.  (suggested by Reviewer p4Wu)

- We fix the citation in Table 1 (suggested by Reviewer SXhF)

- We elaborate on the discussion of relation between CACR, normal CL and CL with hard sample selections. This discussion is extended to the new section 3.3. (suggested by Reviewer co9r)


**Section5: Experiments**

- We rewrite the first paragraph to explain the general settings and clarify some unclear points. (suggested by Reviewer SXhF)

- We add the results of ASCL and ADACLR into Table 4 for comparison  (suggested by Reviewer p4Wu)


- We add the results of full-shot image classification in Table 6 and corresponding discussion (suggested by Reviewer co9r)

*Some typos, and minors pointed out by Review 3T1n are also fixed in the paper. Thanks!*

--------------------------------------------------------

Detailed point-to-point responses to reviewers' comments are provided below.

We hope these revisions address your concerns and we thank you for your time and effort in improving our work.

Sincerely,

Authors

---

### Decision · Action_Editors · 2023-07-05

**Recommendation:** Accept as is

**Comment:**

This paper is reviewed by four experts on this topic, and all the reviewers contributed very good comments and suggestions. We finally collect three final recommendations from the reviewers. All these three reviewers suggested the acceptance, considering that the paper provided enough evidence that support their claims through theory and experiments. In general, this work is interesting enough to be published at TMLR. So the AE suggested the acceptance of this paper.

**Audience:**

This paper should be interested to the researchers of machine learning and computer vision.

**Claims And Evidence:**

This paper presents itself in a form in which it purports to present a “powerful new contrastive leanring method”, but the method is very similar to prior work, and the experimental results support their claims. The authors also provide a theoretical analysis against previous works, which is nice.

This paper is reviewed by four experts on this topic, and all the reviewers contributed very good comments and suggestions. We finally collect three final recommendations from the reviewers. And the AE suggested the acceptance of this paper. This paper is generally well written. There are also some critical defects in the initial submission, and the authours addressed these issues in the revision.

Note that the reviewers still suggest that the paper would have benefitted from being framed as a study on contrastive learning rather than a new powerful method, which should be considered in the camera ready version.